# SFT-GO: Supervised Fine-Tuning with Group Optimization for Large Language Models

## Abstract

Supervised fine-tuning (SFT) has become an essential step in tailoring large language models (LLMs) to align with human expectations and specific downstream tasks. However, existing SFT methods typically treat each training instance as a uniform sequence, giving equal importance to all tokens regardless of their relevance. This overlooks the fact that only a subset of tokens often contains critical, task-specific information. To address this limitation, we introduce Supervised Fine-Tuning with Group Optimization (SFT-GO), a novel approach that treats groups of tokens differently based on their importance. SFT-GO groups tokens in each sample based on their importance values and optimizes the LLM using a weighted combination of the worst-group loss and the standard cross-entropy loss. This mechanism adaptively emphasizes the most challenging token groups and guides the model to better handle different group distributions, thereby improving overall learning dynamics. We provide a theoretical analysis of SFT-GO's convergence rate, demonstrating its efficiency. Empirically, we apply SFT-GO with three different token grouping strategies and show that models trained with SFT-GO consistently outperform baseline approaches across popular LLM benchmarks. These improvements hold across various datasets and base models, demonstrating the robustness and the effectiveness of our method.

## 1 Introduction

Supervised fine-tuning (SFT) plays a crucial role in post-training large language models (LLMs) to better align them with human expectations. This process generally demands a large volume of high-quality data and improves safety, reliability, and suitability for specialized tasks (Radford et al., 2019; Ouyang et al., 2022; Touvron et al., 2023). A key aspect of SFT is to improve data quality and diversity, often via community-driven data collection and synthetic generation (Lambert et al., 2024; Zhou et al., 2023a; Wang et al., 2024b). Recent studies (Li et al., 2024; Zhou et al., 2023a) highlight that effective instruction fine-tuning can be achieved using relatively small but meticulously curated datasets, emphasizing the importance of quality over quantity. LIMA (Zhou et al., 2023a) shows that fine-tuning with 1,030 carefully selected examples can produce high-performing instruction-tuned models, emphasizing the importance of identifying key training signals.

When evaluating data quality, mainstream approaches often focus on individual samples at the instance level, overlooking the fact that tokens within a sample do not contribute equally to task-specific semantics. Many tokens primarily serve functional roles (e.g., conjunctions, articles), while a smaller subset carries critical semantic content that directly impacts performance on downstream tasks. While pretraining equips models with a solid foundation for handling function words, fine-tuning may yield diminishing returns if equal emphasis is placed on all tokens rather than prioritizing those that are semantically rich and task-relevant. Our findings, illustrated in Figure 1, indicate that during supervised fine-tuning (SFT), LLMs exhibit slightly higher cross-entropy (CE) loss when predicting semantically rich tokens, such as "neurons" or "glial cells" (average CE loss: 0.72), compared to more common functional words like "is" or "and" (average CE loss: 0.59). These semantically rich tokens are under-optimized when compared with the common functional tokens.

Translating this motivation into practice, we introduce **Supervised Fine-Tuning with Group Optimization** (SFT-GO). In SFT-GO, tokens are categorized based on their contextual relevance through a designated function, and the model is optimized using a weighted combination of the worst-group

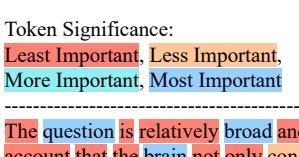

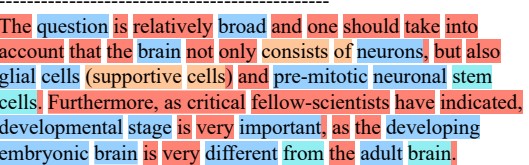

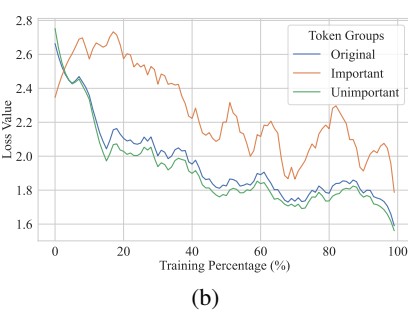

(a)                                    (b)

Figure 1: (a) Text highlighted by LLMLingua-2 based on importance, showing key information from a passage about brain cell migration during development and in adults. (b) Average training loss for models using standard auto-regressive loss. The 'original' line shows overall loss across all tokens, while 'important' and 'unimportant' lines reflect losses for tokens ranked high or low by LLMLingua-2's token compression method.

loss and the standard cross-entropy loss. Inspired by group distributional robust optimization (*Group DRO* (Sagawa* et al., 2020)), this mechanism encourages the model to focus on the group of tokens it currently finds the most difficult, enabling it to better handle diverse token distributions and improving overall learning dynamics. We also prove two theoretical properties: (1) minimizing the SFT-GO objective provides a distributionally robust guarantee that the population worst-group loss is no larger than that obtained by standard CE training, and (2) a simple mini-batch procedure preserves the $O(1/\sqrt{T})$ convergence rate and converges to the global optimum.

A significant advantage of SFT-GO is its flexibility in defining token groups based on various notions of importance. In this paper, we explore three representative strategies to demonstrate this flexibility: (1) a *statistics-based* method using TF-IDF scores, where tokens with higher TF-IDF values are considered more important due to their rarity and relevance within the corpus; (2) a *semantics-based* method that leverages token-selection probabilities from an existing prompt compression model, LLMLingua-2 (Pan et al., 2024), thus incorporating contextual semantic information into the grouping process; and (3) a *task-specific* approach based on an external reference model as in Rho-1 (Lin et al., 2024a), which determines token importance through excess loss calculations and which we demonstrate to be a special case of our proposed framework. These methods focus on different aspects of token importance—statistical significance, contextual relevance, and task-specific utility—highlighting the versatility of SFT-GO in accommodating diverse definitions of token importance, each with its own advantages and limitations.

We evaluated SFT-GO on two widely used instruction-tuning datasets, LIMA (Zhou et al., 2023a) and Alpaca (Taori et al., 2023), using Llama 3.2-3B and Llama 3.1-8B. Our results show that SFT-GO consistently outperforms standard fine-tuning baselines on seven widely recognized benchmarks. Further analysis reveals that the lexical statistics-based grouping strategy, represented by TF-IDF, and the semantic-based approach, represented by LLMLingua-2, yield greater improvements in commonsense reasoning tasks. Our findings demonstrate the promise of token-level optimization and highlight the importance of carefully tailored grouping strategies in instruction-tuning pipelines. The contributions of this paper can be summarized as follows:

- We propose the SFT-GO framework, enabling supervised fine-tuning with group optimization. We provide a mathematical proof for its theoretical convergence.

- We introduce two token grouping methods for SFT-GO, one based on TF-IDF and another relying on compression-based token selection (LLMLingua-2), while reformulating an existing method (Rho-1) within the SFT-GO framework.

- Through empirical experiments, we demonstrate that SFT-GO outperforms baseline SFT approaches.

## 2 RELATED WORK

**Token-Importance** In language models, not all tokens contribute equally; certain words or phrases are more crucial to meaning and performance, and allow the models to focus on the most

relevant information for language tasks (Vaswani, 2017; Jain & Wallace, 2019; Wiegreffe & Pinter, 2019). Recent prompt compression methods, such as LLMLingua (Jiang et al., 2023) and LLMLingua-2 (Pan et al., 2024), have leveraged token importance for efficient inference (Jiang et al., 2024; Mu et al., 2024), compressing less important tokens while preserving semantic integrity during LLM generation. These findings highlight the inherent differences among tokens in terms of their contributions to meaning and task performance. In this paper, we leverage these insights to provide targeted group optimization for instruction fine-tuning, enabling the model to focus on the group that it finds the most difficult to learn during training.

**Leveraging Token-Importance** Previous research in NLP has recognized the significance of incorporating token importance during training (Zaidan et al., 2007; Zhang et al., 2016; Bao et al., 2018). Recently, Gu et al. (2020) assigned weights based on token frequency to improve the performance of language models in machine translation. Similarly, Luo et al. (2023) leveraged token reweighting to address the issue of data imbalance in Named Entity Recognition. The most closely related work in this area is the Rho-1 method proposed by Lin et al. (2024b), which utilizes a reference model fine-tuned on a task-specific dataset to provide signals for token weighting in supervised fine-tuning. In our paper, we propose group optimization as a general instruction fine-tuning framework and establish its theoretical guarantees. We demonstrate that Rho-1 is a special case of our group optimization objective, and we propose two other efficient grouping alternatives that achieve similar performance gains without relying on task-specific fine-tuned reference models.

**Group Optimization** Group Distributionally Robust Optimization has been widely used to address distribution shifts and to enhance model robustness and performance on tasks with different underlying hidden distributions (Hu et al., 2018a;b; Duchi et al., 2019; Shafieezadeh Abadeh et al., 2015; Duchi & Namkoong, 2019). For example, optimizing for the worst-group loss, Sagawa* et al. (2020) shows that the model becomes more fair to minority groups. This technique has also been applied to language modeling for robust performance over different latent topics (Oren et al., 2019). Our paper leverages the theoretical foundation of Group DRO but defines groups in a different way. Instead of treating each data input as a unit, we examine the tokens within each data sample and consider groupings among these tokens. We establish theoretical guarantees under this new setting and substantiate our performance gains through empirical evaluation.

## 3 METHOD

**Question 1.** *Are all tokens equally important?*

To address this question, we first need to define what we mean by *important* tokens. In this paper, we consider important tokens to be a small subset within a sequence that captures the core semantic meaning of the text. There are various empirical methods to evaluate token importance. Traditionally, techniques such as TF-IDF (Term Frequency-Inverse Document Frequency) have been used to identify significant features in text. More recently, prompt compression methods have been utilized to remove redundant information and emphasize essential content (Li et al., 2023).

**Question 2.** *In the context of language modeling, how does the model treat important tokens and unimportant tokens?*

In standard supervised fine-tuning, all tokens are treated equally (Ouyang et al., 2022). The training loss is computed as the average cross-entropy loss for next-token prediction across all tokens. Typically, evaluations focus on this overall average loss. However, when we examine the losses for important and unimportant tokens separately, a significant performance gap emerges.

Figure 1(b) illustrates this phenomenon by comparing the training losses for the important and unimportant token groups. Our observations reveal two key findings:

1. **Early-Stage Proximity**: At the start of training, both groups decrease rapidly, with a visible initial offset between the curves.

2. **Divergence Over Time:** As training progresses, the model maintains a substantial gap between the losses of unimportant and important tokens, consistently achieving lower loss on the unimportant group.

This discrepancy suggests that models disproportionately reduce the loss of less informative tokens, likely due to their high frequency and well-established representations from pretraining, while losses on important tokens remain comparatively harder to reduce. This behavior is undesirable: overemphasizing uninformative words can lead to overfitting, while underfitting on important words hinders the model's ability to capture essential semantic content.

An analogous situation arises in classification tasks with label imbalance (Hu et al., 2018b). Functional words or stylistic elements, which form the basic structure of language, appear frequently and are overrepresented in the training data. In contrast, important semantic tokens are rare and underrepresented. This imbalance in token distribution can negatively affect the model's performance on the minority class, in this case, the important tokens.

In the next section, we will introduce how we employ *group optimization*, a traditional optimization framework, to address this imbalance in data distributions. By adjusting the training objective to focus on the token group that the model handles worst at each step, we ensure that it improves on its weakest subset while still capturing the semantic essence of the text and maintaining overall linguistic coherence, leading to more balanced training dynamics, enhanced robustness against different token distributions, and ultimately stronger generalization. The overview of SFT-GO framework is illustrated in Appendix A.

### 3.1 SUPERVISED FINE-TUNING WITH GROUP OPTIMIZATION

Consider a corpus of $N$ sentences or documents, denoted as $S = s^{(1)}, s^{(2)}, \ldots, s^{(N)}$. Each sentence $s^{(i)}$ is a sequence of tokens:

$$s^{(i)} = [w^{(i,1)}, w^{(i,2)}, \ldots, w^{(i,L_i)}],$$

where $L_i$ is the number of tokens in the $i$-th sentence.

We assume that the importance of each token within a document is evaluated by an grouping function $g(\cdot)$, which assigns a binary label indicating whether a token is important (1) or unimportant (0):

$$g(w^{(i,j)}, s^{(i)}) \in \{1, 0\}. \tag{1}$$

The grouping function determines the importance value of token $w^{(i,j)}$ within its sentence $s^{(i)}$, considering its role in conveying the semantic meaning. The specific choice of $g(\cdot)$ depends on the implementation details and will be discussed in Section 3.3.

Using this assignment, we naturally partition the data into two disjoint groups:

$$G_1 = \left\{ w^{(i,j)} \mid g(w^{(i,j)}, s^{(i)}) = 1 \right\}, G_0 = \left\{ w^{(i,j)} \mid g(w^{(i,j)}, s^{(i)}) = 0 \right\}. \tag{2}$$

where each token occurrence within a sentence belongs exclusively to one of the two groups based on the grouping function $g(\cdot)$.

To optimize the language model $f(\theta)$, we define the group optimization loss function as:

$$\mathcal{L}_{\text{GO}}(w; \theta) = (1 - \lambda) \mathcal{L}_{\text{CE}}(w; \theta) + \lambda \mathcal{L}_{\text{worst}}(w; g, \theta), \tag{3}$$

where:

- $\mathcal{L}_{\text{CE}}(w; \theta)$ is the standard average cross-entropy loss computed over all tokens $w$.
- $\mathcal{L}_{\text{worst}}(w; g, \theta)$ is the *worst-group* loss, defined as:

$$\mathcal{L}_{\text{worst}}(w; g, \theta) = \max(\mathcal{L}_{\text{CE}}(w_{G_1}; \theta), \mathcal{L}_{\text{CE}}(w_{G_0}; \theta)) \tag{4}$$

  where $\mathcal{L}_{\text{CE}}(w_{G_1}; \theta)$ and $\mathcal{L}_{\text{CE}}(w_{G_0}; \theta)$ are the average cross entropy losses computed over the important and unimportant token groups, respectively.
- $\lambda$ is a weighting parameter between 0 and 1 that controls the influence of the group optimization term.

In this objective function, we minimize the standard autoregressive loss while also focusing on reducing the maximum loss between the two groups. By penalizing the worse-performing group,

we encourage the model to perform well across both important and unimportant tokens, ensuring balanced learning.

During implementation, the coefficient $\lambda$ can be either a constant or a decaying function that places more emphasis on group optimization during the early stages of training:

$$\lambda = \max(\lambda_{\max}(1 - t/T), \lambda_{\min}), \tag{5}$$

where we decay the $\lambda$ value from $\lambda_{\max}$ to $\lambda_{\min}$ during the $T$ training steps. This annealing schedule allows the model to focus more on reducing the worst-group loss during the initial training phases and gradually shift emphasis back to the standard loss as training progresses.

## 3.2 THEORETICAL ANALYSIS

We provide a theoretical analysis of SFT-GO and prove that optimizing the group objective $\mathcal{L}_{\mathrm{GO}}$ (Eq. 3) yields better inter-group token balance relative to conventional training approaches. This result is formally stated in the following proposition.

**Proposition 1.** *Let $\hat{\theta}$ be the solution obtained by minimizing the group objective function (Eq. 3), and let $\theta_{avg}$ be the solution obtained by minimizing the standard autoregressive objective function. Then, the worst-group loss of the model trained with group optimization is no greater than that of the model trained with the standard objective:*

$$\mathcal{L}_{\mathrm{worst}}(\hat{\theta}) \leq \mathcal{L}_{\mathrm{worst}}(\theta_{avg}). \tag{6}$$

This proposition provides a population-level, distributionally robust guarantee: under the same train–test distribution, minimizing SFT-GO yields a model whose worst-group expected loss is no larger than that of a CE-trained model. Thus, SFT-GO explicitly improves (or matches) generalization on the group that CE under-optimizes. The detailed proof of Proposition 1 is provided in Appendix B.

Next, we examine the convergence rate of the error in stochastic gradient descent towards the global optimum. We define the excess error after $T$ iterations as:

$$\varepsilon_T := \mathcal{L}_{\mathrm{GO}}(\theta^{(1:T)}) - \min_{\theta \in \Theta} \mathcal{L}_{\mathrm{GO}}(\theta), \tag{7}$$

where $\theta^{(1:T)}$ represents the average of the parameters over steps 1 to $T$. Building upon the foundational work of Sagawa* et al. (2020) on convergence analysis in distributionally robust optimization, we demonstrate that minimizing our proposed optimization function converges at the standard rate of $O(1/\sqrt{T})$. This result is formally presented in the following proposition.

**Proposition 2.** *Suppose that the loss function $\mathcal{L}_{\mathrm{GO}}$ in Eq. 3 is convex and has Lipschitz continuous subgradients, and that the parameter space $\Theta$ is convex, closed, and bounded such that $\|\theta - \theta'\| \leq B_\theta$ for some constant $B_\theta$ for all $\theta, \theta' \in \Theta$. Then, the average parameter $\theta^{(1:T)}$ obtained over $T$ iterations of SGD achieves an expected excess error bounded by:*

$$\mathbb{E}[\varepsilon_T] \leq O\left(1/\sqrt{T}\right), \tag{8}$$

*where the expectation is over the randomness introduced by the sampling in the algorithm.*

The proof of Proposition 2 can be found in Appendix B. This result indicates that our proposed group optimization algorithm is efficient, achieving convergence at the same rate as standard SGD in convex optimization settings.

## 3.3 DEFINING THE GROUPING FUNCTION

SFT-GO is a general framework for incorporating token importance into supervised fine-tuning. The way importance is defined can vary depending on the context. In this section, we explore several options in defining the grouping function $g$ and discuss their pros and cons. Throughout this section, we assume a single hyperparameter $\eta$, which controls the percentage of input tokens that are labeled as important.

**Statistics-Based Grouping**   This approach determines the importance of a token based on its statistical properties within the corpus. For instance, we can define the grouping function $g$ based on token frequency—tokens that are rarer are considered more important than those that appear more often. Alternatively, we can define $g$ based on unigram TF-IDF values: given an input sequence, we group the tokens with the top $\eta$ TF-IDF scores as important, leaving the rest as unimportant. TF-IDF has been widely validated as an effective statistical measure for identifying informative tokens in document-level tasks (Salton & Buckley, 1987), though it does not incorporate contextual information and assigns identical importance to repeated occurrences of the same token within a sequence.

**Semantics-Based Grouping**   Unlike statistics-based methods, this approach utilizes contextual semantics to determine the importance of each token. Here, we consider using a prompt compression method, LLMLingua-2 (Pan et al., 2024), to evaluate the importance of a token. LLMLingua-2 is a BERT-based encoder-only Transformer model. For each token, it generates a probability indicating whether we should keep or drop the token. We group the top $\eta$ tokens as important and treat the rest as unimportant. While this approach incorporates contextual information, a drawback is the requirement of a pre-trained compression model. Fortunately, LLMLingua-2 performs well in general English domains; however, this may not be the case for low-resource languages or specialized domains.

**Loss-Based Grouping**   This approach assigns token importance based on task-specific loss, such as cross-entropy. Tokens with higher losses are considered more important. Rho-1 (Lin et al., 2024a), for example, uses excess loss—the difference between the losses from the current and a fixed reference model:

$$E(w^{(i,j)}) := \mathcal{L}_{\mathrm{CE}}(w^{(i,j)}; \theta) - \mathcal{L}_{\mathrm{CE}}(w^{(i,j)}; \phi), \tag{9}$$

where $\phi$ is a reference model trained on a data from the same domain. Tokens with the highest excess loss (top-$\eta$) are selected. When $\lambda = 1$ in Eq. 3, Rho-1 aligns with our training objective.[1] Compared to the previous two approaches, although loss-based grouping methods are able to integrate the downstream task performance for group selection, this method is often impractical due to the need for a domain-specific reference model.

## 4   EXPERIMENTAL SETUP

**Base Models and Datasets**   We consider two base LLMs for fine-tuning: **Llama-3.2-3B** and **Llama-3.1-8B** (Dubey et al., 2024). For datasets, we utilize two widely used instruction fine-tuning datasets: **LIMA** (Zhou et al., 2023a) and **Alpaca** (Taori et al., 2023). LIMA contains 1,030 high-quality, diverse human-written prompt-response pairs, with 750 pairs carefully selected from community forums and 250 pairs designed manually. In contrast, Alpaca comprises 51,760 synthetic data generated via self-instruct techniques. The two distinct data help evaluate STF-GO's robustness across different data characteristics.

**Grouping Methods**   As discussed in Section 3.3, we consider three different types of grouping strategies for SFT-GO: statistics-based grouping (TF-IDF), semantics-based grouping (LLMLingua-2), and loss-based grouping (Rho-1). We compare these methods with direct supervised fine-tuning (**Base SFT**).

**Benchmarks**   We evaluate the trained models on seven standard benchmarks: **MMLU** (Hendrycks et al., 2021), **MathQA** (Amini et al., 2019), **ARC-C** (Clark et al., 2018), and **OpenBookQA** (Mihaylov et al., 2018), **HellaSwag** (Zellers et al., 2019), **TruthfulQA** (Lin et al., 2022), **IFEval** (Zhou et al., 2023b). Descriptions of each dataset are provided in Appendix D.

---

[1]The grouping function $g$ in Rho-1 relies on the training model to assess the importance of a token. If $g$ continues to change throughout training, the model may fail to converge due to constantly shifting groups. Therefore, we assume that after a certain number of steps, the grouping function becomes deterministic as the training model stabilizes. This phenomenon is observed in practice, as shown in Section 4.

Table 1: The performance of various training methods on standard benchmarks. All methods are fine-tuned on **LIMA** (Zhou et al., 2023a). The 'Avg' column shows the average across the eight columns. In MMLU:X, X indicates the number of shots. For TruthfulQA, we average generation scores and multiple-choice accuracies. For IFEval, we average instruction-level and prompt-level accuracies. Full results for TruthfulQA and IFEval are in Appendix F.

| Method | MMLU:0 | MMLU:5 | MathQA | ARC-C | OpenBookQA | HellaSwag | TruthfulQA | IFEval | Avg. |
|---|---|---|---|---|---|---|---|---|---|
| Llama-3.2-3B | | | | | | | | | |
| Baseline-SFT | 52.78 ± 0.16 | 52.80 ± 0.20 | 32.70 ± 0.32 | 41.72 ± 0.56 | 25.56 ± 2.00 | 57.01 ± 0.17 | 35.29 ± 0.61 | 26.53 ± 0.96 | 40.55 ± 0.29 |
| Rho-1 | 53.10 ± 0.13 | 53.31 ± 0.16 | 33.68 ± 0.66 | 44.08 ± 0.43 | 26.56 ± 1.63 | 57.77 ± 0.09 | 34.08 ± 1.32 | 27.05 ± 1.48 | 41.20 ± 0.25 |
| TF-IDF | 53.55 ± 0.22 | 53.64 ± 0.17 | 32.25 ± 0.60 | 45.39 ± 1.59 | 25.84 ± 1.49 | 59.11 ± 0.22 | 37.65 ± 0.85 | 27.71 ± 1.03 | 41.89 ± 0.32 |
| LLMLingua-2 | 53.05 ± 0.18 | 53.17 ± 0.15 | 32.47 ± 0.50 | 45.44 ± 0.76 | 27.36 ± 1.76 | 59.22 ± 0.28 | 37.33 ± 1.73 | 27.58 ± 1.20 | 41.95 ± 0.33 |
| Llama-3.1-8B | | | | | | | | | |
| Baseline-SFT | 62.05 ± 0.32 | 63.47 ± 0.15 | 39.84 ± 0.26 | 47.75 ± 1.11 | 29.60 ± 0.92 | 57.11 ± 0.26 | 36.16 ± 2.69 | 24.96 ± 1.42 | 45.12 ± 0.23 |
| Rho-1 | 62.30 ± 0.17 | 63.18 ± 0.28 | 39.05 ± 0.53 | 51.45 ± 1.03 | 29.72 ± 2.41 | 61.35 ± 0.26 | 39.81 ± 2.98 | 24.83 ± 1.20 | 46.46 ± 0.54 |
| TF-IDF | 62.05 ± 0.39 | 63.31 ± 0.35 | 37.94 ± 1.09 | 53.24 ± 1.04 | 29.96 ± 1.47 | 63.90 ± 0.25 | 39.06 ± 3.16 | 26.15 ± 1.16 | 46.95 ± 0.70 |
| LLMLingua-2 | 61.63 ± 0.38 | 62.68 ± 0.23 | 39.54 ± 1.05 | 53.28 ± 0.93 | 30.88 ± 1.15 | 64.03 ± 0.34 | 39.41 ± 2.04 | 27.16 ± 1.18 | 47.33 ± 0.52 |

Table 2: The performances of different training methods fine-tuned on the general instruction data, **Alpaca**. The column 'Avg' represents the average performance over the eight columns per method.

| Method | MMLU:0 | MMLU:5 | MathQA | ARC-C | OpenBookQA | HellaSwag | TruthfulQA | IFEval | Avg. |
|---|---|---|---|---|---|---|---|---|---|
| Llama-3.2-3B | | | | | | | | | |
| Baseline-SFT | 54.62 ± 0.17 | 55.83 ± 0.38 | 33.97 ± 0.17 | 46.60 ± 1.82 | 28.92 ± 1.68 | 56.54 ± 0.06 | 44.69 ± 0.81 | 30.47 ± 0.35 | 43.96 ± 0.53 |
| Rho-1 | 55.10 ± 0.12 | 56.70 ± 0.04 | 34.82 ± 0.20 | 45.05 ± 0.32 | 28.60 ± 0.62 | 55.54 ± 0.04 | 49.24 ± 0.51 | 32.65 ± 0.96 | 44.71 ± 0.23 |
| TF-IDF | 55.04 ± 0.21 | 56.42 ± 0.18 | 34.84 ± 0.23 | 45.61 ± 0.37 | 28.12 ± 0.50 | 56.10 ± 0.03 | 50.26 ± 1.43 | 32.37 ± 0.50 | 44.52 ± 0.88 |
| LLMLingua-2 | 54.96 ± 0.04 | 56.55 ± 0.05 | 34.85 ± 0.19 | 45.19 ± 0.27 | 27.60 ± 0.80 | 56.04 ± 0.04 | 51.48 ± 1.47 | 32.47 ± 0.49 | 44.89 ± 0.21 |
| Llama-3.1-8B | | | | | | | | | |
| Baseline-SFT | 63.67 ± 0.07 | 65.63 ± 0.06 | 40.66 ± 0.12 | 56.79 ± 1.32 | 29.16 ± 0.64 | 60.81 ± 0.02 | 43.96 ± 2.26 | 39.51 ± 1.05 | 50.02 ± 0.29 |
| Rho-1 | 64.20 ± 0.20 | 65.73 ± 0.05 | 40.25 ± 0.11 | 56.40 ± 0.21 | 33.28 ± 0.48 | 61.03 ± 0.03 | 49.23 ± 0.85 | 38.65 ± 0.95 | 51.10 ± 0.19 |
| TF-IDF | 64.18 ± 0.29 | 65.72 ± 0.07 | 40.54 ± 0.19 | 55.92 ± 0.42 | 33.04 ± 0.67 | 61.12 ± 0.06 | 50.30 ± 1.57 | 38.32 ± 0.91 | 51.14 ± 0.31 |
| LLMLingua-2 | 63.94 ± 0.18 | 65.65 ± 0.09 | 40.38 ± 0.17 | 55.77 ± 0.13 | 33.04 ± 0.84 | 61.01 ± 0.06 | 51.35 ± 1.58 | 38.58 ± 1.28 | 51.21 ± 0.18 |

**Training Details** For LIMA, we adopt the hyperparameters from the original paper Zhou et al. (2023a): AdamW (Loshchilov & Hutter, 2019) with a learning rate of $1 \times 10^{-5}$, cosine decay, and no warmup. For Alpaca, we search learning rates across methods (Appendix E), training for 1 and 2 epochs on the 3B and 8B models, respectively, with cosine decay, 0.07 warmup, and 0.05 weight decay. For both datasets and model sizes, we fix $\epsilon = 1 \times 10^{-8}$ and $\lambda = 0.9$ (either decayed or static). Results are averaged over five seeds, with full hyperparameter details in Appendix E.

## 5 RESULTS

In this section, we demonstrate the effectiveness of our proposed SFT-GO framework by presenting the following key findings:

- SFT-GO Models Outperform Baseline SFT on average: models trained using the proposed SFT-GO objective outperform the baseline SFT models on both the LIMA and Alpaca datasets under different base LLMs.

- Impact of Grouping Function $g$ on Performance: The choice of grouping function significantly affects performance. In particular, when the grouping function is defined using LLMLingua-2, the models achieve strong performance on general reasoning tasks.

- Ablation Study: We investigate the impact of the important group ratio $\eta$ and the loss weighting factor $\lambda$, demonstrating the robustness of SFT-GO in different settings.

### 5.1 SFT-GO MODELS OUTPERFORM BASELINE SFT

**Performance Comparison on LIMA** Table 1 shows the results for the Llama-3.2-3B and Llama-3.1-8B models fine-tuned on LIMA using different training methods. For both base pre-trained models, all three models trained with SFT-GO (TF-IDF, Rho-1, and LLMLingua-2) outperform the baseline SFT model in terms of average performance. This demonstrates the effectiveness of the proposed group optimization method in SFT. Notably, the semantics-based LLMLingua-2 SFT-GO model achieves scores of 41.95 and 47.33, respectively, significantly surpassing the baseline SFT scores of 40.55 and 45.12.

Table 3: The average scores over the subject-specific benchmarks (SS-B) and the general reasoning benchmarks (GR-B). $\Delta$ indicates the performance improvement of each SFT-GO model over the Baseline-SFT (e.g., $\Delta =$ LLMLingua-2 $-$ Baseline-SFT).

| | LIMA | | | | Alpaca | | | |
| | Llama-3.2-3B | | Llama-3.1-8B | | Llama-3.2-3B | | Llama-3.1-8B | |
| Method | SS-B | GR-B | SS-B | GR-B | SS-B | GR-B | SS-B | GR-B |
|---|---|---|---|---|---|---|---|---|
| BaseSFT | 41.11 | 39.61 | 48.54 | 40.32 | 43.99 | 43.90 | 51.18 | 48.09 |
| Rho-1 | 42.14 | 39.63 | 49.14 | 41.60 | 44.06 | 45.81 | 51.97 | 49.64 |
| $\Delta$ | +1.03 | +0.02 | +0.60 | +1.28 | +0.07 | +1.91 | +0.79 | +1.55 |
| TF-IDF | 42.13 | 41.49 | 49.30 | 42.96 | 44.01 | 46.24 | 51.88 | 49.91 |
| $\Delta$ | +1.02 | +1.88 | +0.76 | +2.64 | +0.02 | +2.34 | +0.70 | +1.82 |
| LLMLingua | 42.30 | 41.38 | 49.60 | 43.24 | 43.83 | 46.66 | 51.75 | 50.31 |
| $\Delta$ | +1.19 | +1.77 | +1.06 | +2.92 | -0.16 | +2.76 | +0.57 | +2.22 |

**Performance Comparison on Alpaca**  Table 2 shows the performance of the Llama-3.2-3B and Llama-3.1-8B models after fine-tuning on the Alpaca instruction dataset. Across both base models, all SFT-GO approaches (TF-IDF, Rho-1, and LLMLingua-2) demonstrate superior average performance compared to the standard SFT baseline. LLMLingua-2, which incorporates semantic guidance, achieves the highest scores of 44.89 and 51.21, respectively, outperforming the baseline values of 43.96 and 50.02.

## 5.2 IMPACT OF GROUPING FUNCTION ON PERFORMANCE

**SFT-GO w/ LLMLingua-2 and TF-IDF Enhance General Reasoning**  SFT-GO models using LLMLingua-2 and TF-IDF show stronger performance on general reasoning benchmarks (HellaSwag, TruthfulQA, IFEval) than on subject-specific tasks (MMLU, MathQA, ARC-C, OpenBookQA).

This improvement for LLMLingua-2 stems from its BERT-based compression method, fine-tuned on online meeting notes (Pan et al., 2024), which better estimates token importance in long-form reasoning than TF-IDF or Rho-1. Meanwhile, TF-IDF effectively highlights rare but informative tokens that often align with key concepts, guiding the model towards answer-relevant cues, making it a strong statistical baseline despite lacking semantic depth.

As shown in Table 3, LLMLingua-2 and TF-IDF with Llama-3.2-3B show the strongest gains on general reasoning (+1.77/+2.76 for LLMLingua-2 and +1.88/+2.34 for TF-IDF). A similar pattern holds for Llama-3.1-8B (+2.92/+2.22 for LLMLingua-2 and +2.64/+1.82 for TF-IDF). This highlights the strength of semantics-based and frequency-based grouping methods in general reasoning tasks.

In contrast, Rho-1 shows more modest improvements in general reasoning. This disparity likely stems from Rho-1's token selection strategy, which prioritizes high-loss tokens during training. Its focus on high-loss tokens may help with knowledge-intensive tasks but is less effective for reasoning tasks that require contextual salience beyond loss-based difficulty.

**Learning Behavior of Different Grouping Methods**  We analyze different training methods by examining Figure 2, which captures the average loss for all tokens ('original'), and for important and unimportant tokens as selected by the grouping function $g$.

Rho-1: Figure 2(a) shows that the unimportant token loss (the solid green curve) remains consistently high throughout training. Rho-1's grouping strategy leverages the "excess error" between the current model and a frozen reference model. The consistently high loss suggests that both the current model and the reference model struggle with these tokens (i.e., both $\mathcal{L}_{\text{CE}}(w; \theta)$ and $\mathcal{L}_{\text{CE}}(w; \phi)$ are high). This indicates that the reference model inadequately captures token importance, hindering effective training. By design, a token $w$ is considered unimportant and is not optimized in the worst-group loss $\mathcal{L}_{\text{worst}}$ if $\mathcal{L}_{\text{CE}}(w; \theta) \approx \mathcal{L}_{\text{CE}}(w; \phi)$.

TF-IDF: Figure 2(b) illustrates that the average loss for important tokens (the dashed orange curve) decreases rapidly and plateaus, while the unimportant token loss (the solid green curve) becomes

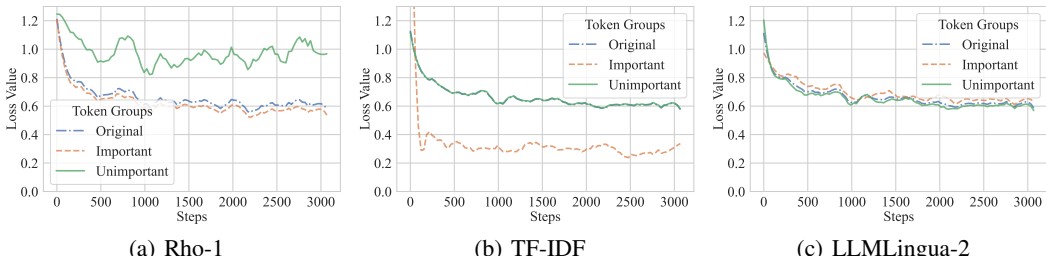

(a) Rho-1          (b) TF-IDF          (c) LLMLingua-2

Figure 2: The progression of losses during model training using the proposed objective function (i.e., group-optimization). The line labeled "original" represents the average loss over all tokens, "important" represents the average loss over important tokens identified by each respective method, and "unimportant" is the average loss over the remaining tokens.

almost identical to the overall loss (the dashed blue curve). TF-IDF's statistics-based approach identifies distinctive lexical patterns but struggles with nuanced semantic relationships. As the important token loss decreases below the unimportant token loss, the worst-group loss $\mathcal{L}_{\text{worst}}$ effectively targets the unimportant token loss

LLMLingua-2 (Figure 2(c)): The loss between important tokens (i.e., the dashed orange curve) and unimportant tokens (i.e., the solid green curve) is relatively small, and both curves closely follow the average loss of all tokens (i.e., the dashed blue curve). This behavior indicates that the worst group loss, $\mathcal{L}_{\text{worst}}$, shifts between the two groups throughout training, focusing on the worst groups as the model progresses. This occurs because the grouping function defined by LLMLingua-2, accurately identifies important tokens by considering the semantics of each token within the context of the document in which it appears.

## 5.3 ABLATION STUDY

**Effect of threshold** $\eta$   In Section 3.3, we discussed the threshold $\eta \in (0, 1)$ used to determine whether a token is important within a document for each grouping function $g$. A higher $\eta$ value results in more tokens being classified as unimportant, effectively serving as the partition threshold. Figure 3 illustrates the average performance across eight benchmarks for different partition thresholds. fine-tuned on LIMA. In both plots, the bars attain the highest values when the compression rate is between 90% and 55%. Across this range, models trained with SFT-GO consistently outperform the baseline-SFT, except for Rho-1 in Llama-3.2-3B at the 90th percentile. This highlights the robustness of SFT-GO.

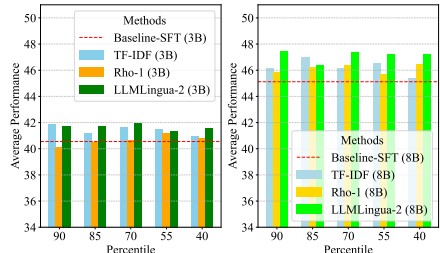

Figure 3: Average performance across benchmarks for Llama-3.2-3B and Llama-3.1-8B trained on LIMA. At the 90th percentile, the bottom 90% of tokens (by $g$) are unimportant; the top 10% are important.

**SFT-GO Outperforms Baseline for Static and Dynamic** $\lambda$   In Section 3.1, we introduced the weight $\lambda$ in Eq. 5, which is a coefficient controlling the impact of worst-group loss in our SFT-GO objective function. By definition, $\lambda$ linearly decays within the range $[\lambda_{min}, \lambda_{max}]$. Notably, $\lambda$ remains static if $\lambda_{min} = \lambda_{max}$. For the decaying case, in this ablation study,

Table 4: Comparison of lambda settings for, LLMLingua and TF-IDF, fine-tuned on LIMA and Alpaca. The column B-SFT represents Baseline-SFT.

| Dataset | Model | Grouping | Static $\lambda$ | Dynamic $\lambda$ | B-SFT |
|---|---|---|---|---|---|
| LIMA | Llama-3.2-3B | LLMLingua | $41.68 \pm 0.86$ | $\mathbf{41.95 \pm 0.33}$ | $40.55 \pm 0.29$ |
| LIMA | Llama-3.2-3B | TF-IDF | $\mathbf{41.89 \pm 0.32}$ | $41.49 \pm 0.48$ | $40.55 \pm 0.29$ |
| Alpaca | Llama-3.2-3B | LLMLingua | $\mathbf{44.89 \pm 0.21}$ | $44.08 \pm 0.17$ | $43.96 \pm 0.53$ |
| Alpaca | Llama-3.2-3B | TF-IDF | $\mathbf{44.52 \pm 0.88}$ | $44.12 \pm 0.57$ | $43.96 \pm 0.53$ |

we set $\lambda_{max} = 0.9$ and $\lambda_{min} = 0.05$ to highlight the contrast between static and dynamic $\lambda$. Table 4 shows that regardless of whether $\lambda$ is static or decaying, SFT-GO using LLMLingua-2 consistently outperforms the baseline-SFT by a large margin. Furthermore, SFT-GO using TF-IDF,

also outperforms the baseline-SFT. This flexibility allows $\lambda$ to be optimized as either a static or a decaying value, and underlines the robustness of the SFT-GO framework.

## 6 CONCLUSION

This paper introduces Supervised Fine-Tuning with Group-Optimization (SFT-GO), a novel approach that enhances LLMs by focusing on different token groups via their importance to model training. SFT-GO groups tokens based on their importance and optimizes the model using a weighted combination of the standard cross-entropy and the worst-group loss. Our theoretical analysis demonstrates the effectiveness of the obtained model on different groups and the efficiency of its convergence rate. Empirical evaluations across multiple token grouping strategies confirm its effectiveness in outperforming the baseline SFT method on standard LLM benchmarks.

## ETHICS STATEMENT

Our work on improving supervised fine-tuning (SFT) methods for large language models has the potential for significant positive societal impact. By enhancing the robustness and generalization of SFT, our approach can make language models more effective across a wide range of applications. However, this method also poses potential risks. Like other LLM training methods, our group-optimization approach may be influenced by biases in the input data, as this work does not specifically address fairness or bias mitigation. One potential solution is to assess and address bias in the data before training.

## REPRODUCIBILITY STATEMENT

This paper presents both experimental and theoretical contributions and provides detailed information to ensure reproducibility. The models used in the experiments are described in Section 4. The training dataset, evaluation benchmarks, and hyperparameters are provided in Appendices E and D. All models and datasets used are open source. For the theoretical contribution, the complete proof is included in Appendix B. A thorough justification of the assumptions is discussed in Appendix C.

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

# A OVERVIEW DIAGRAM

Figure 4 provides an overview of the Supervised Fine-Tuning with Group Optimization (SFT-GO) process.

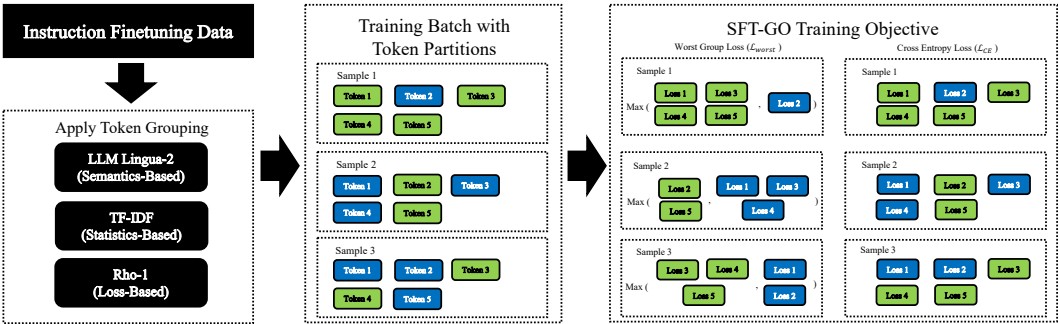

Figure 4: The diagram illustrates the Supervised Fine-Tuning with Group Optimization (SFT-GO) process. Supervised fine-tuning data is processed through token grouping methods: TF-IDF (Statistics-Based), LLMLingua-2 (Semantics-Based), and Rho-1 (Loss-Based). These groups are fed into the SFT-GO objective, which combines standard cross-entropy loss with worst-group loss.

# B PROOF

**Proposition 1.** *Let $\hat{\theta}$ be the solution obtained by minimizing the group objective function (Eq. 3), and let $\theta_{avg}$ be the solution obtained by minimizing the standard autoregressive objective function. Then, the worst-group loss of the model trained with group optimization is no greater than that of the model trained with the standard objective:*

$$\mathcal{L}_{\text{worst}}(\hat{\theta}) \leq \mathcal{L}_{\text{worst}}(\theta_{avg}). \tag{10}$$

*Proof.* For simplicity, prove without the constant $\lambda$ in Eq. 3. Let $\theta_{avg} = \arg\min \mathcal{L}_{\text{CE}}(\theta)$. By definition of $\hat{\theta}$, we obtain

$$\mathcal{L}_{\text{GO}}(w; \hat{\theta}) = (1 - \lambda)\mathcal{L}_{\text{CE}}(\hat{\theta}) + \lambda\mathcal{L}_{\text{worst}}(\hat{\theta}) \tag{11}$$

$$\leq (1 - \lambda)\mathcal{L}_{\text{CE}}(\theta_{avg}) + \lambda\mathcal{L}_{\text{worst}}(\theta_{avg}) \tag{12}$$

$$= \mathcal{L}_{\text{GO}}(\theta_{avg}) \tag{13}$$

for all $\lambda \in (0, 1]$. Suppose

$$\mathcal{L}_{\text{worst}}(\hat{\theta}) > \mathcal{L}_{\text{worst}}(\theta_{avg}). \tag{14}$$

To show

$$(1 - \lambda)\mathcal{L}_{\text{CE}}(\hat{\theta}) + \lambda\mathcal{L}_{\text{worst}}(\hat{\theta}) \leq (1 - \lambda)\mathcal{L}_{\text{CE}}(\theta_{avg}) + \lambda\mathcal{L}_{\text{worst}}(\theta_{avg}), \tag{15}$$

we need $\mathcal{L}_{\text{CE}}(\hat{\theta}) < \mathcal{L}_{\text{CE}}(\theta_{avg})$. However, by definition of $\theta_{avg}$, no other $\theta$ can produce a smaller loss than $\theta_{avg}$ on the loss $\mathcal{L}_{CE}$. Therefore, $\mathcal{L}_{\text{worst}}(\hat{\theta}) > \mathcal{L}_{\text{worst}}(\theta_{avg})$ is not possible. $\square$

**Proposition 2.** *Suppose that the loss function $\mathcal{L}_{\text{GO}}$ in Eq. 3 is convex and has Lipschitz continuous subgradients, and that the parameter space $\Theta$ is convex, closed, and bounded such that $\|\theta - \theta'\| \leq B_\theta$ for some constant $B_\theta$ for all $\theta, \theta' \in \Theta$. Then, the average parameter $\theta^{(1:T)}$ obtained over $T$ iterations of SGD achieves an expected excess error bounded by:*

$$\mathbb{E}[\varepsilon_T] \leq O\left(\frac{1}{\sqrt{T}}\right), \tag{16}$$

*where the expectation is over the randomness introduced by the sampling in the algorithm.*

*Proof.* Assume the followings

- $\Theta$ is convex, closed, and bounded with $||\theta - \theta'|| \leq B_\theta$ for some constant $B_\theta$ and for all $\theta, \theta' \in \Theta$.

- The loss $\mathcal{L}_{GO}$ in Eq. 3 is convex in $\theta$

- The loss $\mathcal{L}_{GO}$ has Lipschitz subgradients (i.e., there exists $G \geq 0$ such as any subgradient $g^{(t)} \in \partial F(\theta)$ satisfies $||g^{(t)}|| \leq G$.

Denote the subgradient of $\mathcal{L}_{GO}$ at iteration $t$ by $g^{(t)}$ and the Euclidean projection onto $\Theta$ by $\prod_\Theta$. The parameter update by the algorithm becomes $\theta^{(t+1)} = \prod_\Theta[\theta^{(t)} - \eta g^{(t)}]$, where $\eta$ is the learning rate.

Denote the solution of $L$ by $\theta^*$ (i.e., $\theta := \arg\min \mathcal{L}_{GO}(\theta)$). Consider the excess error,

$$||\theta^{(t+1)} - \theta^*||^2 \leq ||\theta^{(t)} - \eta g^{(t)} - \theta^*||^2 \tag{17}$$

$$= ||\theta^{(t)} - \theta^*||^2 - 2\eta g^{(t)}(\theta^{(t)} - \theta^*) + \eta^2 ||g^{(t)}||^2. \tag{18}$$

Rearrange to get:

$$g^{(t)}(\theta^{(t)} - \theta^*) \leq \frac{||\theta^{(t)} - \theta^*|| - ||\theta^{(t+1)} - \theta^*||^2}{2\eta} + \frac{\eta}{2} ||g^{(t)}||^2. \tag{19}$$

As $g^{(t)}$ is a subgradient of the convex function $L$, $\mathcal{L}_{GO}(\theta^{(t)}) - \mathcal{L}_{GO}(\theta^*) \leq g^{(t)}(\theta^{(t)} - \theta^*)$. Then we obtain

$$\mathcal{L}_{GO}(\theta^{(t)}) - \mathcal{L}_{GO}(\theta^*) \leq \frac{||\theta^{(t)} - \theta^*|| - ||\theta^{(t+1)} - \theta^*||^2}{2\eta} + \frac{\eta}{2} ||g^{(t)}||^2. \tag{20}$$

Sum the inequality for T steps as:

$$\sum_{t=1}^{T} \left[ \mathcal{L}_{GO}(\theta^{(t)}) - \mathcal{L}_{GO}(\theta^*) \right] \leq \frac{1}{2\eta} \sum_{t=1}^{T} \left( ||\theta^{(t)} - \theta^*|| - ||\theta^{(t+1)} - \theta^*||^2 \right) + \frac{\eta}{2} \sum_{t=1}^{T} ||g^{(t)}||^2. \tag{21}$$

Using telescopic sum, derive

$$\sum_{t=1}^{T} \left( ||\theta^{(t)} - \theta^*|| - ||\theta^{(t+1)} - \theta^*||^2 \right) = ||\theta^{(1)} - \theta^*|| - ||\theta^{(T+1)} - \theta^*||^2 \tag{22}$$

$$\leq ||\theta^{(1)} - \theta^*||^2 \tag{23}$$

$$\leq B_\theta^2. \tag{24}$$

Using this, rewrite Eq. 21 as:

$$\sum_{t=1}^{T} \left[ \mathcal{L}_{GO}(\theta^{(t)}) - \mathcal{L}_{GO}(\theta^*) \right] \leq \frac{B_\theta^2}{2\eta} + \frac{\eta}{2} \sum_{t=1}^{T} ||g^{(t)}||^2. \tag{25}$$

Due to Lipschitz assumption, $\sum_{t=1}^{T} ||g^{(t)}||^2 \leq TG^2$. Therefore,

$$\sum_{t=1}^{T} \left[ \mathcal{L}_{GO}(\theta^{(t)}) - \mathcal{L}_{GO}(\theta^*) \right] \leq \frac{B_\theta^2}{2\eta} + \frac{\eta}{2} TG^2. \tag{26}$$

Choose learning rate $\eta = D/G\sqrt{T}$ and plug it in Eq. 26. Then we obtain

$$\sum_{t=1}^{T} \left[ \mathcal{L}_{GO}(\theta^{(t)}) - \mathcal{L}_{GO}(\theta^*) \right] \leq B_\theta G\sqrt{T}. \tag{27}$$

For the average iterate $\bar{\theta}^{(1:T)} = \sum_{t=1}^{T} \theta^{(t)}/T$, Jensen's inequality gives

$$\mathcal{L}_{GO}(\bar{\theta}^{(1:T)}) \leq \frac{1}{T} \sum_{t=1}^{T} \mathcal{L}_{GO}(\theta^{(t)}). \tag{28}$$

Subtract $\mathcal{L}_{\text{GO}}(\theta^*)$ from Eq. 28 and combine with Eq. 27, then

$$\mathcal{L}_{\text{GO}}(\bar{\theta}^{(1:T)}) - \mathcal{L}_{\text{GO}}(\theta^*) \leq \frac{1}{T} \sum_{t=1}^{T} \left[ \mathcal{L}_{\text{GO}}(\theta^{(t)}) - \mathcal{L}_{\text{GO}}(\theta^*) \right] \tag{29}$$

$$\leq B_\theta G / \sqrt{T} \tag{30}$$

Thus, $E[\varepsilon_T] \leq O(1/\sqrt{T})$. $\qquad\square$

## C    JUSTIFICATION OF THEORETICAL ASSUMPTION

In this section, we explain why each assumption in Proof B—Lipschitz subgradients, bounded domains, and convexity—is not merely a technical convenience but is instead aligned with common preprocessing practices in modern neural network training.

- Bounded-domain: In practice, one almost always enforces an effective norm bound on network parameters via techniques such as weight-decay or clipping steps. This ensures that $||\theta - \theta'||$ is bounded, exactly as our first assumption requires.

- Lipschitz continuous subgradients for $L_{GO}$: The cross-entropy term $L_{CE}$ over a bounded parameter region has bounded gradients, provided inputs and logits are bounded. In practice, inputs are bounded due to rescaling or standardization and logits are also bounded due to techniques such as weight-decay or clipping. Taking a maximum of two such Lipschitz-gradient functions preserve the property. Thus, if $||\nabla L_{CE}|| \leq G$, then any subgradient of $L_{GO}$ also satisfies $||\partial L_{GO}|| \leq G$.

- Convexity: This is a widely accepted assumption in DRO (Ben-Tal et al., 2013; Duchi et al., 2021; Bertsimas et al., 2018; Sagawa* et al., 2020). Much of the fundamental theoretical work (Sagawa* et al., 2020) in *distributionally robust optimization* (DRO), including the convergence guarantees used in our analysis, is based on classical results from convex optimization, such as online mirror descent and results from Nemirovski et al. (2009). Additionally, the DRO methods have been successfully applied to deep neural networks in prior empirical studies (Jiang et al., 2022; Mustapha et al., 2023), suggesting that the theoretical insights from convex analysis can still offer useful guidance. Similarly, Proposition 2 serves as a proof-of-concept that our algorithm operates within a stable and well-behaved regime.

## D    DESCRIPTION OF THE BENCHMARKS

- **MMLU** (Hendrycks et al., 2021) (0-shot and 5-shot) is a comprehensive benchmark testing model on broad knowledge and reasoning in 57 different subjects.

- **MathQA** (Amini et al., 2019) (0-shot) consists of math word problems that require the extraction of key information from natural language narratives and the conversion of the narratives into executable meaning representations.

- **ARC-C** (Clark et al., 2018) tests the multi-step reasoning of models via science questions that cannot be solved with just retrieval.

- **OpenBookQA** (Clark et al., 2018) evaluates a model's ability to combine a fact with reasoning to answer science questions.

- **HellaSwag** (Zellers et al., 2019) (0-shot) tests on broad knowledge and reasoning in 57 diverse subjects.

- **TruthfulQA** (Lin et al., 2022) assesses a model's ability to provide factually accurate answers, even when questions are designed to elicit common misconceptions or falsehoods.

- **IFEval** (Zhou et al., 2023b) evaluates a model's natural language instruction following ability.

# E  HYPER-PARAMETERS

## E.1  LIMA

For experiments on LIMA (Zhou et al., 2023a), we follow the hyperparameters recommended in the original paper. For both Llama-3.2-3B and Llama-3.1-8B, we train for 15 epochs using the AdamW optimizer (Loshchilov & Hutter, 2019) with a learning rate of $1e-5$, a weight decay of 0.1, an epsilon value of $1e-8$. We use a cosine annealing schedule without any warm-up steps, a batch size of 64, and a maximum context window size of 4096.

For the implemented SFT-GO methods, we tuned the compression rate $\eta$ and the coefficient $\lambda$. The selected hyperparameters are:

- For TF-IDF, we use an $\eta$ value of the 90th percentile for Llama-3.2-3B and the 85th percentile for Llama-3.1-8B. For $\lambda$, we use a static value of 0.9 for both models.

- For Rho-1, we use $\eta$ values of 55th percentile and 70th percentile for Llama-3.2-3B and Llama-3.1-8B, respectively. Rho-1 does not have a tunable $\lambda$.

- For LLMLingua-2, we found the best $\eta$ to be the 70th percentile for both models. For $\lambda$, we use either a decaying value from 0.9 to 0.07 for Llama-3.2-3B and a static value of 0.9 for Llama-3.1-8B.

## E.2  ALPACA

For all experiments on Alpaca (Taori et al., 2023), we train using the following protocol. For both the Llama-3.2-3B and Llama-3.1-8B models, we use the AdamW optimizer (Loshchilov & Hutter, 2019) with an epsilon value of $1e-8$ and cosine annealing schedule. We use a batch size of 32 and a maximum context window of 4096.

For Llama-3.2-3B, we train for 1 epoch with a warm-up ratio of 0.07 and a weight decay of 0.05. We search the optimal learning rate, $\eta$, and $\lambda$. The selected hyperparameters are:

- For the baseline-SFT, the learning rate of $6e-6$ is used.

- For TF-IDF, we use an $\eta$ value of the 70th percentile, static $\lambda$ of 0.9, and learning rate of $1e-6$.

- For Rho-1, we use an $\eta$ value of the 70th percentile and a learning rate of $5e-7$. Rho-1 does not use $\lambda$ by construction.

- For LLMLingua-2, we found the best $\eta$ to be the 25th percentile. For $\lambda$, we use a static value of 0.9 and learning rate of $6e-7$.

For Llama-3.1-8B, we train for 2 epochs with a warm-up ratio of 0.07 and a weight decay of 0.05. The baseline model uses a best learning rate of $1e-7$, and the SFT-GO models use a learning rate of $6e-7$. The learning rates for both the baseline and the SFT-GO models follow a cosine annealing schedule. We tune the compression rate $\eta$ and the coefficient $\lambda$. The selected hyperparameters are:

- For TF-IDF, we use an $\eta$ value of the 60th percentile. For $\lambda$, we use a static value of 0.9.

- For Rho-1, we use an $\eta$ value of the 40th percentile. Rho-1 does not have a tunable $\lambda$.

- For LLMLingua-2, we found the best $\eta$ to be the 25th percentile. For $\lambda$, we use a decaying value from 0.9 to 0.05.

# F  DETAILED TRUTHFULQA & IFEVAL SCORES

Tables 5, 6, 7, and 8 are the in-depth TruthfulQA and IFEval scores for the different training methods mentioned in Table 1 and Table 2.

Table 5: The performances of different training methods on the TruthfulQA benchmarks. All methods are fine-tuned on the general instruction data, **LIMA**. The column 'Avg' represents the average performance over the six sub-benchmarks in TruthfulQA.

| Method | Base | BLEU | ROUGE1 | ROUGE2 | ROUGEL | MC1 | MC2 | Avg. |
|---|---|---|---|---|---|---|---|---|
| Baseline-SFT | Llama-3.2-3B | 36.64 ± 0.65 | 36.79 ± 1.50 | 29.86 ± 0.69 | 36.16 ± 1.37 | 27.78 ± 0.65 | 44.52 ± 0.76 | 35.29 ± 0.61 |
| Rho-1 | Llama-3.2-3B | 35.96 ± 1.28 | 35.23 ± 2.45 | 28.64 ± 1.96 | 34.42 ± 2.22 | 26.83 ± 0.72 | 43.42 ± 0.48 | 34.08 ± 1.32 |
| TF-IDF | Llama-3.2-3B | 40.32 ± 1.55 | 39.68 ± 0.98 | 33.46 ± 2.05 | 37.94 ± 0.76 | 28.91 ± 0.46 | 45.57 ± 0.33 | 37.65 ± 0.85 |
| LLMLingua-2 | Llama-3.2-3B | 40.56 ± 2.13 | 39.14 ± 2.59 | 32.39 ± 2.83 | 38.70 ± 2.37 | 28.42 ± 0.66 | 44.77 ± 0.63 | 37.33 ± 1.73 |
| Baseline-SFT | Llama-3.1-8B | 37.72 ± 3.31 | 41.67 ± 3.49 | 21.30 ± 5.79 | 40.22 ± 3.12 | 29.30 ± 1.66 | 46.75 ± 1.53 | 36.16 ± 2.69 |
| Rho-1 | Llama-3.1-8B | 41.62 ± 4.30 | 46.78 ± 8.80 | 30.26 ± 4.71 | 44.21 ± 9.57 | 30.11 ± 1.12 | 45.89 ± 1.47 | 39.81 ± 2.98 |
| TF-IDF | Llama-3.1-8B | 40.61 ± 4.45 | 44.62 ± 6.42 | 28.95 ± 1.82 | 42.84 ± 6.22 | 29.10 ± 1.91 | 45.34 ± 3.14 | 39.06 ± 3.16 |
| LLMLingua-2 | Llama-3.1-8B | 42.03 ± 3.80 | 44.67 ± 3.78 | 31.51 ± 0.68 | 43.35 ± 3.91 | 29.15 ± 1.27 | 45.75 ± 1.06 | 39.41 ± 2.04 |

Table 6: The performances of different training methods on the IFEval benchmarks. All methods are fine-tuned on the general instruction data, **LIMA**. The word 'Inst' refers to instruction, and the word 'Acc' refers to accuracy. The column 'Avg' represents the average performance over the four sub-benchmarks in IFEval.

| Method | Base | Inst Level Loose Acc | Inst Level Strict Acc | Prompt Level Loose Acc | Prompt Level Strict Acc | Avg. |
|---|---|---|---|---|---|---|
| Baseline-SFT | Llama-3.2-3B | 34.00 ± 0.79 | 33.04 ± 0.82 | 20.04 ± 1.26 | 19.04 ± 1.27 | 26.53 ± 0.96 |
| Rho-1 | Llama-3.2-3B | 34.17 ± 1.44 | 33.45 ± 1.27 | 20.67 ± 1.80 | 19.93 ± 1.59 | 27.05 ± 1.48 |
| TF-IDF | Llama-3.2-3B | 35.23 ± 0.84 | 34.15 ± 0.86 | 21.22 ± 1.46 | 20.26 ± 1.21 | 27.71 ± 1.03 |
| LLMLingua-2 | Llama-3.2-3B | 34.96 ± 1.73 | 34.10 ± 1.60 | 21.00 ± 0.87 | 20.26 ± 0.87 | 27.58 ± 1.20 |
| Baseline-SFT | Llama-3.1-8B | 32.40 ± 2.53 | 31.39 ± 1.40 | 18.45 ± 1.66 | 17.60 ± 1.41 | 24.96 ± 1.42 |
| Rho-1 | Llama-3.1-8B | 32.16 ± 1.26 | 30.98 ± 1.25 | 18.56 ± 1.20 | 17.64 ± 1.16 | 24.83 ± 1.20 |
| TF-IDF | Llama-3.1-8B | 33.00 ± 1.29 | 32.31 ± 1.32 | 19.55 ± 1.23 | 18.81 ± 1.14 | 26.15 ± 1.16 |
| LLMLingua-2 | Llama-3.1-8B | 34.58 ± 1.06 | 33.76 ± 1.07 | 20.55 ± 1.48 | 19.74 ± 1.22 | 27.16 ± 1.18 |

Table 7: The performances of different training methods on the TruthfulQA benchmarks. All methods are fine-tuned on the general instruction data, **Alpaca**. The column 'Avg' represents the average performance over the six sub-benchmarks in TruthfulQA.

| Method | Base | BLEU | ROUGE1 | ROUGE2 | ROUGEL | MC1 | MC2 | Avg. |
|---|---|---|---|---|---|---|---|---|
| Baseline-SFT | Llama-3.2-3B | 47.03 ± 1.68 | 52.93 ± 1.75 | 30.04 ± 0.67 | 52.39 ± 1.89 | 34.03 ± 0.45 | 51.76 ± 0.11 | 44.69 ± 0.81 |
| Rho-1 | Llama-3.2-3B | 55.50 ± 1.18 | 59.41 ± 0.69 | 33.44 ± 2.06 | 59.76 ± 0.91 | 34.86 ± 0.28 | 52.49 ± 0.24 | 49.24 ± 0.51 |
| TF-IDF | Llama-3.2-3B | 55.52 ± 1.73 | 62.42 ± 2.29 | 34.56 ± 2.47 | 62.47 ± 2.69 | 34.58 ± 0.26 | 51.72 ± 0.24 | 50.26 ± 1.43 |
| LLMLingua-2 | Llama-3.2-3B | 57.21 ± 2.12 | 64.72 ± 2.61 | 32.73 ± 1.69 | 66.17 ± 4.34 | 35.46 ± 0.34 | 52.58 ± 0.24 | 51.48 ± 1.47 |
| Baseline-SFT | Llama-3.1-8B | 45.19 ± 3.39 | 47.03 ± 2.29 | 35.98 ± 5.66 | 44.95 ± 2.14 | 37.28 ± 0.28 | 53.31 ± 0.24 | 43.96 ± 2.26 |
| Rho-1 | Llama-3.1-8B | 51.82 ± 1.49 | 57.92 ± 1.79 | 33.51 ± 1.03 | 57.45 ± 2.00 | 39.07 ± 0.36 | 55.62 ± 0.14 | 49.23 ± 0.85 |
| TF-IDF | Llama-3.1-8B | 53.93 ± 2.51 | 59.14 ± 2.33 | 35.28 ± 2.70 | 59.41 ± 3.36 | 38.83 ± 0.20 | 55.19 ± 0.29 | 50.30 ± 1.57 |
| LLMLingua-2 | Llama-3.1-8B | 55.32 ± 2.42 | 60.17 ± 1.73 | 37.55 ± 4.19 | 61.47 ± 2.81 | 38.70 ± 0.34 | 54.89 ± 0.17 | 51.35 ± 1.58 |

Table 8: The performances of different training methods on the IFEval benchmarks. All methods are fine-tuned on the general instruction data, **Alpaca**. The word 'Inst' refers to instruction, and the word 'Acc' refers to accuracy. The column 'Avg' represents the average performance over the four sub-benchmarks in IFEval.

| Method | Base | Inst Level Loose Acc | Inst Level Strict Acc | Prompt Level Loose Acc | Prompt Level Strict Acc | Avg. |
|---|---|---|---|---|---|---|
| Baseline-SFT | Llama-3.2-3B | 37.58 ± 0.53 | 36.57 ± 0.67 | 24.44 ± 0.27 | 23.29 ± 0.37 | 30.47 ± 0.35 |
| Rho-1 | Llama-3.2-3B | 40.19 ± 0.82 | 38.95 ± 0.77 | 26.32 ± 1.31 | 25.14 ± 1.17 | 32.65 ± 0.96 |
| TF-IDF | Llama-3.2-3B | 38.87 ± 0.46 | 37.94 ± 0.55 | 26.84 ± 0.63 | 25.84 ± 0.66 | 32.37 ± 0.50 |
| LLMLingua-2 | Llama-3.2-3B | 39.57 ± 0.56 | 38.61 ± 0.52 | 26.40 ± 0.68 | 25.32 ± 0.47 | 32.47 ± 0.49 |
| Baseline-SFT | Llama-3.1-8B | 48.10 ± 1.24 | 44.53 ± 0.86 | 34.71 ± 1.26 | 30.68 ± 0.96 | 39.51 ± 1.05 |
| Rho-1 | Llama-3.1-8B | 46.59 ± 0.83 | 45.46 ± 0.77 | 31.79 ± 1.13 | 30.76 ± 1.15 | 38.65 ± 0.95 |
| TF-IDF | Llama-3.1-8B | 45.92 ± 0.94 | 44.94 ± 0.90 | 31.61 ± 1.03 | 30.83 ± 0.96 | 38.32 ± 0.91 |
| LLMLingua-2 | Llama-3.1-8B | 46.06 ± 1.43 | 45.15 ± 1.43 | 32.05 ± 1.21 | 31.05 ± 1.16 | 38.58 ± 1.28 |

# G    TOKEN SPLITTING BASELINE

We examine the effectiveness of different grouping strategies against the random grouping baseline, where the tokens are randomly split into two groups.

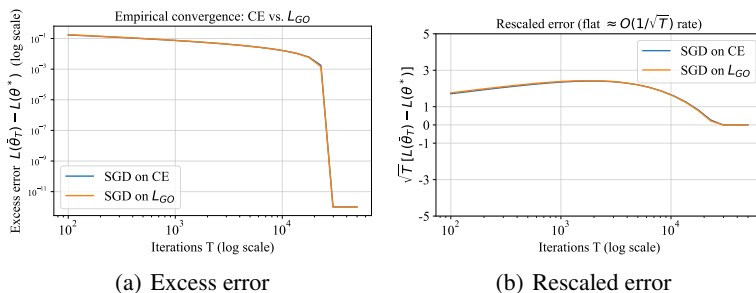

(a) Excess error            (b) Rescaled error

Figure 5: Convergence behavior of minibatch SGD on the standard cross-entropy objective (CE) and the proposed group-optimization objective $L_{GO}$ in a convex logistic-regression setting.

Table 9 reports the average performance across methods. We observe that the random grouping strategy performs on par with Baseline-SFT. In contrast, all SFT-GO variants outperform random grouping. We hypothesize that this is because random grouping does not introduce any bias toward token importance.

Table 9: Ablation of grouping methods for Llama-3.2-3B fine-tuned on LIMA. The values report average performance across eight evaluation benchmarks.

|  | Baseline-SFT | Random Group | Rho-1 | TF-IDF | LLMLingua-2 |
|---|---|---|---|---|---|
| Avg. | $40.55 \pm 0.29$ | $40.97 \pm 0.24$ | $41.20 \pm 0.25$ | $41.89 \pm 0.32$ | $\mathbf{41.95 \pm 0.33}$ |

## H  EMPIRICAL VALIDATION OF CONVERGENCE RATE

To empirically validate Proposition 2, we constructed a fully convex logistic-regression setting with a fixed grouping function and ran minibatch SGD. Figure 5(a) plots the excess error $L(\bar{\theta}_T) - L(\theta^*)$ in log-log scale. The CE and $L_{GO}$ curves almost perfectly overlap and decay along a straight line, consistent with the expected $1/\sqrt{T}$ rate. Figure 5(b) plots the rescaled quantity $\sqrt{T}[L(\bar{\theta}_T) - L(\theta^*)]$; both methods produce nearly flat curves over the entire optimization horizon, which is the characteristic signature of $O(1/\sqrt{T})$ convergence. These results empirically confirm the convergence rate of Proposition 2 in a convex regime.

## I  DISCUSSION ON SAMPLING METHODS

Sample-level data valuation methods assign scalar values to the training examples, identifying which samples are more influential or informative for downstream generalization (Choe et al., 2024; Wang et al., 2024a). They operate at sequence granularity. In contrast, SFT-GO adjusts optimization within each sequence using token-level priors. These two methods target distinct failure modes: (1) Sample-level methods determine which examples to train on, and (2) SFT-GO determines how optimization is allocated within each token group.

Table 10: Comparison of the performances obtained by the sampling method and by LLMLingua-2 combined with the complementary sampling method.

| Method | Score |
|---|---|
| Baseline_SFT | 40.55 |
| LESS (Xia et al., 2024) | 41.36 |
| LLMLingua-2 | 41.95 |
| LLMLingua-2 + LESS | 42.46 |

Because these approaches are orthogonal, combining them may provide additive gains. We incorporated a representative and widely used sample-selection baseline: LESS (Xia et al., 2024), a probabilistic data valuation method that selects training examples according to learned sample importance.

We conducted experiments on the LIMA dataset using Llama-3.2-3B, applying LESS to select the top 40% highest-value samples. As shown in Table 10, the sampling method LESS (Xia et al., 2024) is effective and outperforms the Baseline-SFT when fine-tuned on the selected data. LLMLingua-2, however, still surpasses LESS. Furthermore, combining the token-prior method with the complementary sampling strategy yields even better performance.

## J   ANALYSIS OF TOKEN IMPORTANCE ASSIGNMENT BY EACH GROUPING FUNCTION

In this section, we analyze and compare three distinct methodologies — LLMLingua-2, TF-IDF, and Rho-1—for estimating token importance in textual data. Each method is grounded in a different underlying principle, leading to notable divergences in token selection criteria and interpretability.

LLMLingua-2 assigns token importance based on token probability, where words with lower predictive confidence are deemed more informative. This probabilistic approach inherently aligns with syntactic structure and textual coherence, as uncertainty often arises at transitional points in a sentence or phrase. Consequently, as shown in Figure 6, connective elements such as "but," "although," and "as long as" receive higher importance, suggesting that LLM Lingua emphasizes tokens that mediate sentence flow and grammatical dependencies. This approach highlights words that govern linguistic cohesion and sentence progression, ensuring that the model accurately predicts discourse-level fluency rather than merely content-bearing terms.

In contrast, TF-IDF operates purely on statistical frequency, striking a balance between its frequency within a given corpus and its rarity This methodology is entirely independent of semantic context or model-based representations, leading to a prioritization scheme that is lexically, rather than semantically or syntactically, driven. In Figure 7, tokens such as "human", "stored" and "not intended" do receive the highest weight, yet function words and syntactic markers such as "but" ,"and", and "the" comprise the majority. This underscores a fundamental limitation of TF-IDF: it overemphasizes conjunction words regardless of their actual semantic contribution, potentially amplifying domain-specific jargon while failing to distinguish between truly informative and merely infrequent terms. While this property makes TF-IDF effective for retrieval-based tasks such as keyword extraction, its lack of linguistic sensitivity renders it suboptimal for applications requiring semantic interpretation or syntactic coherence.

Finally, the Rho-1 method assigns token importance based on excess loss, capturing tokens that exhibit a disproportionately high contribution to model uncertainty. Unlike LLMLingua, which prioritizes syntactic fluency, and TF-IDF, which emphasizes rarity, Rho-1 systematically identifies tokens that are inherently difficult for the model to predict. As a result as shown in Figure 8, it preferentially assigns high importance to key domain-specific terms, such as "buns," "heated," and "temperature," while deprioritizing syntactic connectors and predictable function words. This distinction suggests that Rho-1 operates as a semantically-aligned importance estimator, favoring words that directly impact model learning and downstream task performance.

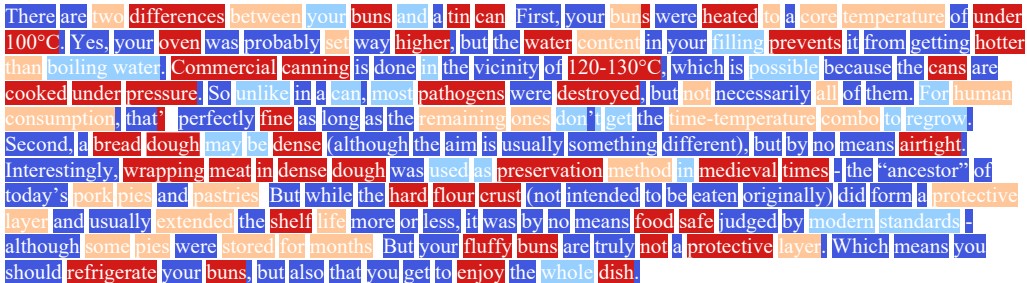

Figure 6: Token importance visualization based on the LLMLingua. Importance is assigned based on syntactic flow and function words. Dark blue indicates higher importance, while dark red represents lower importance.

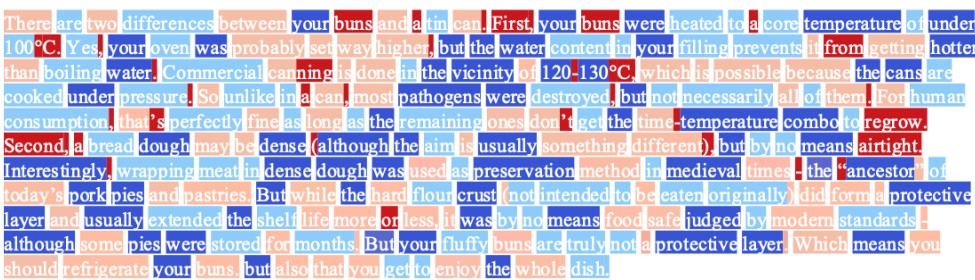

Figure 7: Token importance visualization based on the TF-IDF. This approach highlights rare terms without considering context or syntactic structure.

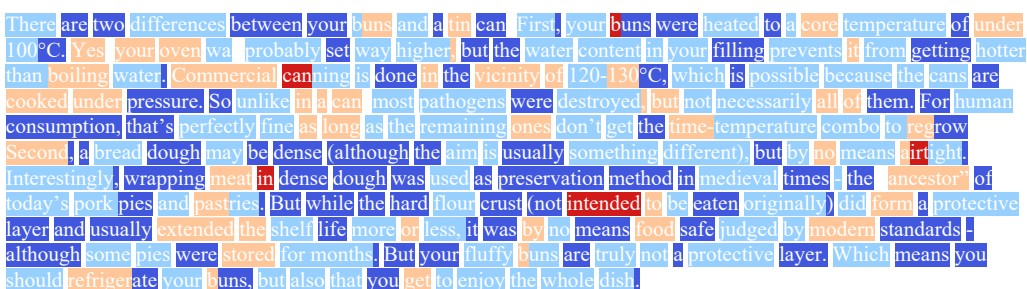

Figure 8: Token importance visualization based on Rho-1. This method assigns importance based on training difficulty, emphasizing semantically challenging tokens.

## K   GENERATED OUTPUT BY DIFFERENT TRAINING METHODS AND ANALYSIS

This section presents example outputs from our models and analyzes their characteristics. The examples below demonstrate how different training methods affect the style, content, and quality of responses when models are given the same user query about planning a trip to Japan.

When comparing the outputs from different training methods, we observe distinct patterns in response structure, content depth, and recommendation style:

- **Baseline-SFT** produces a moderately detailed response with conversational elements, but includes some potentially inaccurate information (e.g., Kyoto Tower being in Tokyo).

- **TF-IDF Evaluator** generates the most comprehensive response with specific recommendations for locations, activities, and even restaurants. The response includes more factual details but may be overwhelming in its level of specificity.

- **Rho-1 Evaluator** offers the most concise response, focusing on high-level planning strategy rather than specific attractions. It emphasizes experiencing both urban and rural Japan but provides fewer concrete recommendations.

- **LLMLingua-2 Evaluator** delivers a well-structured, bullet-point itinerary with a balance of specific attractions and broader cultural context. The response is organized by day and includes diverse activities across multiple cities.

---

**Llama-3.2-3B fine-tuned on LIMA via Baseline-SFT**

**User:** I am planning a trip to Japan for three days, and I would like you to write an itinerary for my journey.
**Assistant:** The diversity of Japan's culture and geography means that the "perfect" three-day itinerary is likely to differ depending on the interests of the traveller. That said, there are a few core experiences which typically figure highly in travellers' lists of recommendations - so much so, that they can often be found in competition with one another at certain locations (Tokyo's sushi vs Osaka's ten-maki!). With no particular order other than ascending price-point, this is a Tokyo-focused three-day plan which hits two of the country's most-famous icons:
Day 1: * Start in Shinjuku and spend the morning floating around Kyoto Tower photoshoot HQ before moving on to nearby Akihabara. * Finish off Day 1 in Ginza, either by transferring to another part of your plan or to dinner with views of al fresco alconomers at Tokyo Station's Cucumber Market.
Day 2: * Head to the imperial capital for the day by switching railways at Niigata; options including Uga Creek and Kawakita Bonodori-en include an evening destination. * Alternatively, stay in Tokyo and focus on trying to beat your neighbour to ten-maki at one of three identified hubs - Roppongi, Shinjuku & New Tokyo, Akasfair & Ikebukuro - each featuring pros and cons.
And finally,
Day 3: * Bus or train trips like the one to Niseko above are common, as is combining them with a ski lesson. * Depending on timing, a Hiroshima/Tokayama trio may be missed; alternatives and alternatives to Oita exist.

---

## Llama-3.2-3B fine-tuned on LIMA via TF-IDF Evaluator

**User:** I am planning a trip to Japan for three days, and I would like you to write an itinerary for my journey.

**Assistant:** Here is a suggested three-day itinerary for your trip to Japan:

Day 1: Your starting point might be Tokyo, which is well-known as a modern metropolis with a mix of traditional and innovative culture. You could consider visiting some famous landmarks such as Tokyo Tower, the Sky Emporium in Roppongi, or the Tokyo City View observation deck located on the 52nd floor of the Almont Hotel. For those who want to experience Japanese traditions and cultures, there are plenty options too! The Asakusa district features a historic red-brick town hall dating back to the Meiji period (around the same time as our country's first parliament building), and a beautiful Sensoji Temple surrounded by quaint shops selling local specialties like mochi and Kit-Kat milk chocolate buns designed exclusively for overseas visitors. Another popular spot among locals & tourists alike is Ueno Park -one of the most cherished parks in Tokyo- where you'll find over a hundred cherry trees blossoming beautifully during May-June each year if you happen to visit at the right time! To cap off Day One, head over to Shinjuku at dusk when all the lamps come alive; this bustling business hub also boasts the world's tallest free-standing structure – the Tokyo Metropolitan Government Building. On its observatory floors, you can take in stunning panoramas of the city lights below. End your day by treating yourself to sumptuous French cuisine at Gagome, a Michelin-starred restaurant serving up delectable 10-course kaiseki meals prepared using locally-sourced ingredients.

Day 2: For Day Two, consider basing yourself in Kyoto instead of moving on from Tokyo. Often referred-to as Japan's cultural capital, Kyoto is home to more than 100 temples and shrines that together make up half of the country's World Heritage Site designation! Some must-visit spots here include Kinkakuji (Golden Pavilion), Ritsurin Park, and the bamboo forest at Arashiyama. In the evening, head out to Osaka for dinner and catch the nightly Hokkaido Ramen Tour that departs from Nipponbashi Pier around 5pm. This unique experience allows you to savor authentic Hokkaido ramen while enjoying breathtaking views of illuminated Tokyo towerlit against the night sky. Be sure to book in advance as slots are limited!

Day 3: On your final day, why not fly from Itami Airport in Osaka to Naha on the island of Okinawa? This destination is often overlooked but is actually one of my favorite places in Japan. It boasts gorgeous beaches, crystal-clear water for snorkeling and scuba diving, delicious seafood cuisine (try the spicy tuna bowl at Menya Kodai!), hot springs (onsens) without the crowds found elsewhere in Japan, and plenty of other unique attractions worth experiencing before flying back home through Osaka or Kansai International Airports.

---

**Llama-3.2-3B fine-tuned on LIMA via Rho-1 Evaluator**

**User:** I am planning a trip to Japan for three days, and I would like you to write an itinerary for my journey.
**Assistant:** For three-day trips, my recommendation is to pick two regions in Japan that you want to see and spend an equal amount of time on each. This way, you can experience the culture of Japan in its truest form, without rushing from place to place.
Example:
Day 1 & 2 - Tokyo
Day 1 - Osaka/Kyoto
Day 3 - Leave from a rail station in either Tokyo or Osaka and visit Hiroshima/Nagasaki
In this example, you will be able to experience Japanese life from start (cities) to finish (rural areas). The rural area portion of your trip will help you to better understand why Japan has been ranked #1 in tourism over the last several years. If there is a specific location you are interested in seeing, feel free to adjust the plan accordingly!

---

**Llama-3.2-3B fine-tuned on LIMA via LLMLingua-2 Evaluator**

**User:** I am planning a trip to Japan for three days, and I would like you to write an itinerary for my journey.
**Assistant:** Japan is a beautiful country that offers something unique to visitors from all over the world. The country's combination of modern and traditional cultures will leave any tourist in awe. While in Tokyo, Kyoto, and Hiroshima, make sure to try out some authentic Japanese cuisine, go on a shopping spree in search of high-quality souvenirs, and most importantly, experience first-hand how different Japanese culture really is.
Day 1: * Visit the Imperial Residence in East Tokyo to view the changing of the guard at noon. * Take a stroll through Shinjuku's bustling streets to see modern Japanese life. * Dine at Ichigoyukata, one of the top restaurants in the world, for a memorable and luxurious meal.
Day 2: * Explore Akihabara, often referred to as the "Electric Appliance Town of Tochigi," to find some great gadgets and collectibles. * Spend your afternoon in Kyoto, widely considered the seventh capital of Japan. * Travel to Nara during nightfall and into the evening to enjoy this city's stunning illuminated scenery.
Day 3: * Head to the National Museum of Emerging Sciences in Okazaki to learn about cutting-edge technology. * Tour Huis Ten Bosch, a theme park inspired by European architecture. * Spend some time exploring Hiroshima Peace Memorial Park and Atomic Bomb Dome to reflect on the past.
As you can see, each day takes visitors to different parts of Japan - Tokyo, Kyoto, and Hiroshima - while also allowing them to experience the true essence of Japanese culture. This way, they'll get a better understanding of life in Japan as a whole. Try your best to visit Japan sometime soon!

