# OpenReview forum: "SFT-GO: Supervised Fine-Tuning with Group Optimization for Large Language Models"
_ICLR.cc/2026/Conference — Submitted to ICLR 2026_

### Official Review · Reviewer_r4vC · 2025-10-21

**Soundness:** 2
**Presentation:** 3
**Contribution:** 2
**Rating:** 2
**Confidence:** 4

**Summary:**

This paper studied the token importance for supervised fine-tuning in large language models. It was motivated by the observation that important and unimportant tokens had different average CE losses during fine-tuning. To improve supervised fine-tuning, this work introduced a novel SFT-GO approach by grouping tokens based on their importance values and then optimizing LLMs using a weighted combination of the worst-group loss and the standard cross-entropy loss. Experimental results demonstrated that the proposed approach outperformed standard supervised fine-tuning in large language models.

**Strengths:**

**Originality:** The paper was motivated by the empirical results in Figure 1 that important and unimportant tokens had different average CE losses during supervised fine-tuning in LLMs. It then introduced a novel SFT-GO approach based on the group distributional robust optimization.

**Quality:** In addition to empirical validation, the convergence of the proposed approach was theoretically analyzed. It showed that Rho-1 (Lin et al., 2024a) could be considered as a special case of the proposed framework. Experimental results demonstrated that the proposed approach outperformed standard supervised fine-tuning in large language models. The learning behavior of different grouping methods was analyzed.

**Clarity:** The paper is well-written. Most of the proposed techniques in SFT-GO are easy to follow. The experimental settings are presented in the submission (as well as the appendix).

**Significance:** The proposed approach provides a generic alternative for supervised fine-tuning in LLMs. It is also flexible to extend the proposed approach by considering more advanced token importance estimation strategies in various downstream tasks.

**Weaknesses:**

**W1:** The motivation of this paper is not well justified and explained.
- Figure 1(b) shows the average CE losses of important and unimportance tokens. It states that "at the beginning of training, the losses for both groups are similar". But it seems that important and unimportance tokens have significantly different loss curves at the beginning of training in Figure 1(b).
- Following the previous concern, section 3 shows the key findings of Figure 1(b) that the losses for the two groups start similarly but diverge over time. However, the decay function in Eq. (5) allows the model to "focus more on reducing the worst-group loss during the initial training phases and gradually shift emphasis back to standard loss as training processes". This seems to be contradictory. Since the losses diverge later, it might make more sense to focus on the worst-group loss more as training goes on, not less.
- Line 162-163 states that models focus on reducing the loss of less informative tokens, possibly because they are easier to optimize. It is confusing why the loss of less informative tokens is easier to optimize.

**W2:** The theoretical analysis of SFT-GO is unconvincing.
- Proposition 1 compares the performance of the worst-performing group using SFT-GO and standard supervised fine-tuning. However, the proof in Appendix B shows that "for simplicity, prove without the constant $\lambda$ in Eq. 3." It is unclear whether Proposition 1 holds for any value of $\lambda$.
- It seems that Proposition 1 focuses only on the training loss over the worst-performing group. The generalizability of fine-tuned models using SFT-GO and standard supervised fine-tuning is still unclear.
- Lines 73-74 mention the theoretical properties that "minimizing our objective improves the accuracy of the model in all token groups compared to standard training". This theoretical property is not justified.

**W3:** The baseline comparison and hyperparameter sensitivity can be further discussed.
- Lines 296-297 show that when $\lambda=1$ in Eq. 3, Rho-1 aligns with the proposed training objective (the appendix also mentions that Rho-1 does not have a tunable $\lambda$). But it is unclear whether the experiments consider Rho-1 as one baseline approach in existing work. If so, it seems that the proposed approaches based on TF-IDF and LLMLingua-2 have marginal performance improvement compared to the existing Rho-1 baseline.
- The selection of the hyperparameter $\lambda$ is not discussed. It is unclear why $\lambda=0.9$ is selected for all settings.

**Questions:**

(1) Proposition 2 discusses the convergence rate of the proposed approach. However, compared to traditional DRO, this work focuses on the supervised fine-tuning process, by initializing the parameters using a pre-trained LLM. In contrast, traditional DRO will train a neural network from scratch. Will the different initialization approaches affect the convergence rate? More specifically, will pre-trained LLM enable improving the convergence rate compared to models with random initialization?

(2) Lines 274-275 show that "TF-IDF has proven to be a robust statistical measure for token importance". This proof can be further validated, especially when it also mentions that TF-IDF "does not account for contextual information".

---

> ### Author Response · Authors · 2025-11-25
> **Rebuttal by Authors**
>
> > Weakness 1 (W1): Figure 1(b) shows the average CE losses of important and unimportance tokens. It states that "at the beginning of training, the losses for both groups are similar". But it seems that important and unimportance tokens have significantly different loss curves at the beginning of training in Figure 1(b).
>
> Thank you for pointing this out. Our wording was imprecise. In Fig. 1(b), there is an initial offset between the important and unimportant loss curves at the very start. We have updated the manuscript to reflect this more accurately.
>
> We intended to show that the key phenomenon is the persistent separation that develops over training, where the unimportant group consistently attains lower loss while the important group remains higher, under the standard autoregrssive loss objective. We have revised the paper to clarify.
>
>
> > W2: Following the previous concern, section 3 shows the key findings of Figure 1(b) that the losses for the two groups start similarly but diverge over time. However, the decay function in Eq. (5) allows the model to "focus more on reducing the worst-group loss during the initial training phases and gradually shift emphasis back to standard loss as training processes". This seems to be contradictory. Since the losses diverge later, it might make more sense to focus on the worst-group loss more as training goes on, not less.
>
> Following your suggestion, we tested an increasing lambda schedule (0.01 → 0.9) and found that it led to lower accuracy (40.71) compared to our original decaying schedule (41.95). This supports the effectiveness of decaying lambda for group optimization. As shown in Figure 1b, group distinctions become more evident later in training, so emphasizing group optimization early helps the model learn more balanced representations. Notably, all SFT-GO models still outperformed the baseline-SFT (40.55), confirming the robustness of our method.
>
> > W3: Line 162-163 states that models focus on reducing the loss of less informative tokens, possibly because they are easier to optimize. It is confusing why the loss of less informative tokens is easier to optimize.
>
> Thank you for the comment. Our intent was not to claim that less informative tokens are inherently easier, but that they empirically exhibit lower loss and faster reduction during SFT. Frequent functional tokens (e.g., "is," "and") are already well-learned during pretraining and have low-variance predictive distributions, so their CE loss decreases quickly. In contrast, semantically rich tokens are rarer and more context-dependent, leading to higher initial loss and slower optimization. We revised the main paper to clarify it.
>
> > W4: Proposition 1 compares the performance of the worst-performing group using SFT-GO and standard supervised fine-tuning. However, the proof in Appendix B shows that "for simplicity, prove without the constant $\lambda$ in Eq. 3." It is unclear whether Proposition 1 holds for any value of $\lambda$.
>
> Thank you for the helpful comment. The argument extends directly when the weighting parameter is reinstated. The key observation is that $\lambda$ merely scales (and does not change the ordering of) the worst-group loss term. Thus, the minimizer of the SFT-GO objective still cannot have a strictly larger worst-group loss than the minimizer of the standard CE objective. We have revised the proof to make this explicit by keeping $\lambda$ throughout. Please refer to Appendix B in the revised paper.
>
> > W5: It seems that Proposition 1 focuses only on the training loss over the worst-performing group. The generalizability of fine-tuned models using SFT-GO and standard supervised fine-tuning is still unclear.
>
> The inequality in Proposition 1 is a population-level generalization guarantee: under the same train-test distribution, minimizing SFT-GO yields a model whose worst-group expected loss is no larger than that of a model trained with standard CE.
>
> The purpose of Proposition 1 is not to show improvements on arbitrary downstream metrics, but to formalize the specific failure mode that motivates SFT-GO: the standard CE systematically under-optimizes semantically important token groups. By guaranteeing a strictly better (or equal) worst-group population risk, the proposition provides a distributionally robust rationale for emphasizing these under-optimized tokens. We have clarified this implication in the revised main paper, and we invite the reviewer to refer to the updated discussion following Proposition 1.

---

> ### Author Response · Authors · 2025-11-25
> **Rebuttal by Authors**
>
> > W6: Lines 73-74 mention the theoretical properties that "minimizing our objective improves the accuracy of the model in all token groups compared to standard training". This theoretical property is not justified.
>
> Thank you for reading it carefully. The statement in Lines 73–74 referred to Proposition 1, and we agree that the sentence was misleading. Proposition 1 establishes an inequality on the worst-group population loss, showing that minimizing SFT-GO yields a model whose worst-group expected loss is no larger than that of a CE-trained model under the same train–test distribution.
>
> We have revised the text in the main paper to more accurately reflect the theoretical result. The corrected statement now reads: "Minimizing the SFT-GO objective provides a distributionally robust guarantee that the population worst-group loss is no larger than that obtained by standard CE training."
>
> > W7: Lines 296-297 show that when $\lambda=1$ in Eq. 3, Rho-1 aligns with the proposed training objective (the appendix also mentions that Rho-1 does not have a tunable $\lambda$). But it is unclear whether the experiments consider Rho-1 as one baseline approach in existing work. If so, it seems that the proposed approaches based on TF-IDF and LLMLingua-2 have marginal performance improvement compared to the existing Rho-1 baseline.
>
> We would like to reiterate that SFT-GO is a general framework for supervised fine-tuning with group optimization. As discussed in Section 1 (Lines 99–101) and formalized in Lines 296–297, Rho-1 corresponds to a special case of SFT-GO obtained by setting λ=1 and using the excess-loss grouping function (Eq. 9). Our goal in reformulating Rho-1 within the SFT-GO framework is not to present it as a new method, but to show that existing heuristics for prioritizing certain token groups can be unified and better understood under a principled group-optimization formulation.
>
> In our extensive experiments, the results show that (i) all SFT-GO variants outperform standard CE training, and (ii) the choice of token-grouping strategy meaningfully affects performance: different grouping rules yield different gains across subject-specific and general-reasoning benchmarks (Table 3). Beyond these empirical differences, the SFT-GO formulation provides theoretical grounding and generality, enabling the use of alternative grouping mechanisms tailored to task characteristics.
>
> > W8: The selection of the hyperparameter $\lambda$ is not discussed. It is unclear why $\lambda=0.9$ is selected for all settings.
>
> We performed an extensive initial grid search for $\lambda$ when choosing its value. Regardless of the $\lambda$ value, the SFT-GO methods outpeform the baseline SFT. We reproduce some of the variations of the lambda values in the table below for Llama-3.2-3B on LIMA.
>
>
> | Metric         | Model            | $\lambda$     | LIMA     |
> |----------------|------------------|------------|------------|
> | Base-SFT        | Llama-3.2-3B     | N/A | 40.55 ± 0.290   |
> | LLMLingua-2        | Llama-3.2-3B     | 0.3 | 41.63 ± 0.770 |
> | LLMLingua-2      | Llama-3.2-3B     | 0.7| 41.79 ± 0.150 |
>
> The $\lambda = 0.9$ setting (41.95 ± 0.33) yields the best performance, but all $\lambda$ variants outperform the base SFT, demonstrating the robustness of the method.
>
>
> > Question 1 (Q1): Proposition 2 discusses the convergence rate of the proposed approach. However, compared to traditional DRO, this work focuses on the supervised fine-tuning process, by initializing the parameters using a pre-trained LLM. In contrast, traditional DRO will train a neural network from scratch. Will the different initialization approaches affect the convergence rate? More specifically, will pre-trained LLM enable improving the convergence rate compared to models with random initialization?
>
> Proposition 2 provides a worst-case asymptotic convergence guarantee for SGD. The convergence rate does not depend on how the parameter $\theta^{(0)}$ is initialized; random initialization or pretrained initialization yields the same rate. A pretrained LLM may reduce the distance to an optimum and thus improve the constant and early-phase convergence in practice, but it does not improve the theoretical rate. Our proposition is intended as an optimization-stability guarantee for the objective rather than a claim that pretraining alters asymptotic SGD rates.
>
> > Q2: Lines 274-275 show that "TF-IDF has proven to be a robust statistical measure for token importance". This proof can be further validated, especially when it also mentions that TF-IDF "does not account for contextual information".
>
> Thank you for the comment. Our intention in describing TF-IDF as a "robust statistical measure for token importance" was to reference its empirical effectiveness in information retrieval and document understanding tasks [1]. We revised the paper to clarify.
>
> [1] Term-weighting approaches in automatic text retrieval

---

> > ### Comment · Reviewer_r4vC · 2025-11-25
> >
> > Thanks for providing the rebuttal. Here are some follow-up concerns.
> >
> > (1) What does "population-level generalization guarantee" mean in Proposition 1? Does the worst-group loss in both Eq. (4) and Eq. (4) correspond to the expected loss w.r.t. the entire data distribution? If so, are $\hat{\theta}$ and $\theta_{avg}$ assumed to be optimized under the entire data distribution rather than finite training samples?
> >
> > (2) It states that "frequent functional tokens (e.g., "is," "and") are already **well-learned during pretraining** and have low-variance predictive distributions, so their CE loss decreases quickly. In contrast, semantically rich tokens are rarer and more context-dependent, leading to **higher initial loss** and slower optimization." However, based on the results in Figure 1(b), it seems. that frequent functional tokens (unimportant tokens) have much larger loss values than semantically rich tokens (important tokens) at the beginning of training. This phenomenon seems to be inconsistent with your explanations.

---

> > > ### Author Response · Authors · 2025-11-27
> > > **Rebuttal by Authors**
> > >
> > > > What does "population-level generalization guarantee" mean in Proposition 1? Does the worst-group loss in both Eq. (4) and Eq. (4) correspond to the expected loss w.r.t. the entire data distribution? If so, are $\hat{\theta}$ and $\theta_{avg}$ assumed to be optimized under the entire data distribution rather than finite training samples?
> > >
> > > Thank you for the prompt response. By "population-level generalization guarantee," we mean that Proposition 1 is stated in terms of the expected population under the underlying data distribution, including worst-group loss defined as an expectation over each group’s subpopulation. In theory, $\hat{\theta}$ and $\theta_{avg}$ denote population minimizers of their respective objectives; in practice, they are approximated by minimizing empirical objectives on finite samples. This population-vs-empirical setup is standard in DRO/group DRO analyses and their generalization guarantees [1,2,3].
> > >
> > >
> > > [1] Shiori Sagawa, Pang Wei Koh, Tatsunori B. Hashimoto, Percy Liang. Distributionally Robust Neural Networks for Group Shifts: On the Importance of Regularization for Worst-Case Generalization.
> > >
> > > [2] Hongseok Namkoong, John C. Duchi. Stochastic Gradient Methods for Distributionally Robust Optimization with f-Divergences.
> > >
> > > [3] John C. Duchi, Hongseok Namkoong. Learning Models with Uniform Performance via Distributionally Robust Optimization.
> > >
> > > > It states that "frequent functional tokens (e.g., "is," "and") are already well-learned during pretraining and have low-variance predictive distributions, so their CE loss decreases quickly. In contrast, semantically rich tokens are rarer and more context-dependent, leading to higher initial loss and slower optimization." However, based on the results in Figure 1(b), it seems. that frequent functional tokens (unimportant tokens) have much larger loss values than semantically rich tokens (important tokens) at the beginning of training. This phenomenon seems to be inconsistent with your explanations.
> > >
> > > Thank you for the clarification request. Our statement that semantically rich tokens have "higher initial loss and slower optimization" refers to their behavior in the early optimization regime, not the exact ordering at the first iteration. Our intended point is about the optimization dynamics: despite any initial offset, the unimportant group’s loss decreases more quickly, while the important group remains harder to reduce and stays higher throughout training.

---

### Official Review · Reviewer_Ha2G · 2025-10-31

**Soundness:** 2
**Presentation:** 3
**Contribution:** 2
**Rating:** 4
**Confidence:** 4

**Summary:**

The paper introduces SFT-GO, a supervised fine-tuning objective that augments standard CE with a worst-group token loss, where “important” vs “unimportant” tokens are pre-selected by a separate grouping/importance heuristic. The method is simple to implement and is evaluated by fine-tuning LLaMA-3.2-3B and LLaMA-3.1-8B on LIMA/Alpaca, reporting modest average gains across several standard benchmarks. The paper also includes a convergence discussion under abstract smoothness/convexity assumptions.

**Strengths:**

1. Clear and simple idea: the objective is easy to reproduce and plug into an existing SFT pipeline; training-time hyperparameters are straightforward (importance threshold, mixing weight, optional schedule).
2. Ablations exist: the paper varies the importance threshold and the mixing schedule and shows that the method does not collapse under reasonable ranges.
3. Reproducibility: datasets, backbones, and overall training setup are sufficiently specified; the method does not require invasive code changes.

**Weaknesses:**

1. Empirical impact is marginal: reported gains are small (often ~1–2 points or within noise), with several tasks showing negligible or no improvement. The paper does not present statistical significance or per-task confidence intervals, so it is hard to assess robustness.
2. Limited scope of models and data: only two relatively small LLaMA variants are tested, both within the same family and on narrow English instruction data. There are no results on larger backbones, multilingual settings, or human judgments—limiting external validity.
3. Unconvincing motivation figures: It is not convincing to treat Fig. 1(b) as the motivation, although "the loss for more informative, higher-value tokens remains relatively stable", the loss for important token groups declines consistently, and the figure does not show the convergence of the loss curve.
4. Theory is largely generic: the convergence argument hinges on broad assumptions under which many losses would also converge; the paper does not actually prove that the proposed ($L_{GO}$) satisfies these assumptions in the non-convex LM setting, nor does it connect the assumptions to practical training dynamics.
5. Missing justification vs CE: beyond intuition, there is no theoretical or diagnostic evidence (e.g., calibration, gradient-conflict, or error-type analyses) explaining *why* emphasizing worst-group tokens should outperform CE on downstream metrics. The preliminary plots do not make this case.
6. Potential implementation ambiguities: the objective contains a (\max) over token groups; the paper lacks detail on gradient flow/stability near ties, and on how frequently the “worst” group switches during training.

**Questions:**

1. Scaling & generality: Do the gains persist for larger backbones and different model families?
2. Objective mechanics: In practice, does the gradient back-propagate only through the currently worse token group, or through both groups? How are ties handled, and how stable are updates when the arg-max group flips frequently?
3. Assumptions vs reality: Can you prove that ($L_{GO}$) satisfies the assumptions used in your convergence discussion? If not, can you provide empirical diagnostics linking those assumptions to observed training behavior?
4. Why better than CE? Beyond intuition, can you show analyses (e.g., error types, per-token calibration, gradient interference, or learning-curve segments) that explain when/why ($L_{GO}$) helps—and when it does not?

---

> ### Author Response · Authors · 2025-11-25
> **Rebuttal by Authors**
>
> > Weakness 1 (W1): Empirical impact is marginal: reported gains are small (often ~1–2 points or within noise), with several tasks showing negligible or no improvement. The paper does not present statistical significance or per-task confidence intervals, so it is hard to assess robustness.
>
> Based on the suggestion, we additionally performed statistical significance testing with the null hypothesis that there is no difference in the average mean performance across benchmarks between the SFT-GO method (e.g., LLMLingua-2) and the baseline SFT. The alternative hypothesis is that the SFT-GO method outperforms the baseline SFT.
>
> We report the p-values in the table below. 11 out of 12 experiments show statistically significant values (p < 0.05). The p-value results indicate significant performance improvements for the SFT-GO models over the Baseline-SFT across all experiments.
>
> P-values. Statistically significant values (p < 0.05) are **bolded**.
>
> | Metric         | Model            | LIMA       | Alpaca     |
> |----------------|------------------|------------|------------|
> | TF-IDF         | Llama-3.2-3B     | **0.0001** | 0.1324     |
> | Rho            | Llama-3.2-3B     | **0.0027** | **0.0152** |
> | LLMLingua      | Llama-3.2-3B     | **0.0001** | **0.0068** |
> | TF-IDF         | Llama-3.1-8B     | **0.0014** | **0.0002** |
> | Rho            | Llama-3.1-8B     | **0.0015** | **0.0001** |
> | LLMLingua      | Llama-3.1-8B     | **0.0001** | **0.0001** |
>
> > W2: Limited scope of models and data: only two relatively small LLaMA variants are tested, both within the same family and on narrow English instruction data. There are no results on larger backbones, multilingual settings, or human judgments—limiting external validity.
>
> Following your suggestion, we conducted additional experiments using various model sizes from a different model family: Qwen-2.5-1.5B and Qwen-2.5-14B. Here is the performance of the SFT-GO variant LLMLingua on the Qwen-2.5 model family, on the LIMA dataset. As shown in the table, SFT-GO with minimal tuning, outperforms the baseline-SFT models.
>
> | Metric         | Model            | LIMA       |
> |----------------|------------------|------------
> | Base-SFT         | Qwen-2.5-1.5B    | 42.23 ± 0.160 |
> | LLMLingua        | Qwen-2.5-1.5B    | 43.12 ± 0.03 |
> | Base-SFT    | Qwen-2.5-14B     | 55.83 ± 0.104 |
> | LLMLingua        | Qwen-2.5-14B      | 56.45 ± 0.001 |
>
> We restrict our experiments to English instruction data to obtain a well-controlled setting and to enable easier comparison with existing works, even though SFT-GO itself is language-agnostic.
>
> > W3: Unconvincing motivation figures: It is not convincing to treat Fig. 1(b) as the motivation, although "the loss for more informative, higher-value tokens remains relatively stable", the loss for important token groups declines consistently, and the figure does not show the convergence of the loss curve.
>
> Fig. 1(b) is intended as qualitative motivation. It illustrates that under the standard autoregressive objective, optimization progress is often stronger on less informative tokens, while losses on important tokens remain comparatively harder to reduce, creating a persistent gap that motivates SFT-GO’s balancing objective. We thank the reviewer, and have updated the manuscript text accordingly, removing the “relatively stable” phrasing.
>
> The figure is not meant to characterize end-stage training behavior. Separately, our theoretical analysis provides formal guarantees for optimizing the SFT-GO objective, including a convergence bound under standard assumptions explored in our response to W4.

---

> ### Author Response · Authors · 2025-11-25
> **Rebuttal by Authors**
>
> > W4: Theory is largely generic: the convergence argument hinges on broad assumptions under which many losses would also converge; the paper does not actually prove that the proposed ($L_{GO}$) satisfies these assumptions in the non-convex LM setting, nor does it connect the assumptions to practical training dynamics.
>
> These conditions are widely used in the theoretical machine learning literature [1, 2, 3]. They are not merely technical conveniences but align with common preprocessing practices in modern neural network training. We included full justification of the theoretical assumptions in Appendix C of the original paper; here, we provide a summary. Please refer to Appendix C for additional details.
>
> - **Bounded-domain:** Practical techniques such as weight decay and parameter clipping keep neural network parameters within a bounded region. This ensures that the distance $\| \theta - \theta' \|$ remains bounded, satisfying the first assumption.
> - **Lipschitz continuous subgradients for $L_{GO}$:** If the inputs and logits remain bounded, which is typically ensured in practice through normalization, weight decay, or clipping, then the cross-entropy loss $L_{CE}$ has uniformly bounded gradients, i.e., $\|\nabla L_{CE}\| \leq G$. The proposed objective $L_{GO}$, which is the pointwise maximum of two Lipschitz-gradient functions, preserves the property. Therefore, every subgradient of $L_{GO}$ satisfies $\|\partial L_{GO}\| \leq G$. This justifies the assumption in practical settings.
> - **Convexity assumption:** Convexity is a standard assumption in DRO and establishes core theoretical guarantees based on classical convex optimization [1, 2, 3, 4]. Empirical applications to deep neural networks show that these insights remain practically useful. Within this context, Proposition 2 further demonstrates that the proposed algorithm functions in a stable, well-behaved regime.
>
> [1] Robust solutions of optimization problems affected by uncertain probabilities
> [2] Statistics of robust optimization: A generalized empirical likelihood approach
> [3] Data-driven robust optimization
> [4] Distributionally robust neural networks
>
> > W5: Missing justification vs CE: beyond intuition, there is no theoretical or diagnostic evidence (e.g., calibration, gradient-conflict, or error-type analyses) explaining why emphasizing worst-group tokens should outperform CE on downstream metrics. The preliminary plots do not make this case.
>
> Our method SFT-GO is motivated by the intuition that tokens serve different roles: some are primarily functional, while others carry the core semantic meaning. As discussed in Section 3, several empirical observations support this perspective. For example, TF–IDF highlights informative tokens using frequency-based signals, and token-compression techniques [5] show that many tokens can be removed without degrading model performance. Our analysis in Figure 1 further illustrates this disparity: semantically rich tokens (e.g., "neurons," "glial cells") incur higher average cross-entropy loss (0.72), whereas common functional words (e.g., "is," "and") incur substantially lower loss (0.59) when trained with standard CE.
>
> Proposition 1 formalizes this imbalance between groups. It guarantees that minimizing SFT-GO yields a model whose worst-group loss is no larger than that of a CE-trained model. Thus, SFT-GO provides a distributionally robust correction to CE by explicitly optimizing the maximum group loss and preventing optimization pressure from concentrating on the easy group.
>
> Finally, extensive experiments confirm that all SFT-GO variants consistently outperform standard CE, demonstrating the effectiveness of addressing the group-level imbalance.
>
> [5] Compressing context to enhance inference efficiency of large language models

---

> ### Author Response · Authors · 2025-11-25
> **Rebuttal by Authors**
>
> > W6: Potential implementation ambiguities: the objective contains a (\max) over token groups; the paper lacks detail on gradient flow/stability near ties, and on how frequently the “worst” group switches during training.
>
> We use PyTorch in our implementation. By construction, when the losses of two groups are exactly equal, PyTorch’s max operator assigns a gradient of 0.5 to each group, causing them to contribute equally to the update. In practice, this almost never occurs, since it is unlikely for the loss values of two groups to be exactly identical.
>
> We show the learning behavior of both token groups and different grouping methods in Figure 2. Below is a brief summary of the analysis. We have revised the analysis in Section 5.2 for clarity; please refer to that section for details:
>
> Across grouping strategies, loss behaviors differ notably. In Rho-1, the unimportant-token loss remains high because both current and reference models struggle, showing that this reference-based method inadequately captures token importance. With TF-IDF, the important-token loss falls quickly but then plateaus, while the unimportant-token loss soon tracks the overall loss and becomes almost identical; this reflects TF-IDF’s reliance on lexical statistics, which ignore semantic relationships and cause optimization to revert to standard training. By contrast, LLMLingua-2 keeps important, unimportant, and original curves closely aligned, with the worst-group loss shifting dynamically between different groups; this indicates its semantics-aware grouping more effectively identifies and balances important tokens during training.
>
> > Question 1 (Q1): Scaling & generality: Do the gains persist for larger backbones and different model families?
>
> Please refer to our response to W2.
>
> > Q2: Objective mechanics: In practice, does the gradient back-propagate only through the currently worse token group, or through both groups? How are ties handled, and how stable are updates when the arg-max group flips frequently?
>
> Please refer to our response to W6.
>
> > Q3: Assumptions vs reality: Can you prove that ($L_{GO}$) satisfies the assumptions used in your convergence discussion? If not, can you provide empirical diagnostics linking those assumptions to observed training behavior?
>
> We rely on three assumptions in Proposition 2: a bounded domain, Lipschitz-continuous subgradients for $L_{GO}$, and convexity. As discussed in Appendix C, the bounded-domain and Lipschitz-subgradient conditions can be empirically verified in practice, while the convexity assumption is widely adopted in DRO. In our setting, it serves as a proof-of-concept simplification that enables a stable and well-behaved theoretical analysis despite the underlying non-convexity of neural networks.
>
> We provide empirical diagnostics showing that the bounded-domain and bounded-subgradient assumptions hold in our training regime. For this, we validate using the SFT-GO variant, LLMLingua. With weight decay (and gradient clipping), the global $L_2$ norm stayed within [1135.27, 1135.38], indicating optimization occurs in a compact region. We additionally cheked gradient norms of $L_{GO}$, which remained bounded and spike-free (max global norm 13.17), supporting the Lipschitz-subgradient condition.
>
> > Q4: Why better than CE? Beyond intuition, can you show analyses (e.g., error types, per-token calibration, gradient interference, or learning-curve segments) that explain when/why ($L_{GO}$) helps—and when it does not?
>
> Thank you for the comment. As discussed in our response to W5, Proposition 1 shows an analysis why $L_{GO}$ helps by establishing that minimizing $L_{GO}$ provably reduces the population worst-group loss relative to CE, independent of specific failure modes such as calibration or gradient conflict.
>
> Because $L_{GO}$ relies on a token-prior, the choice of grouping function influences where the method provides the most benefit. Section 5.2 and Table 3 illustrate this clearly: all SFT-GO variants outperform the CE baseline across both general-reasoning and subject-specific evaluations, but different grouping strategies excel in different regimes (e.g., LLMLingua-2 and TF-IDF yield larger gains on general reasoning, whereas Rho-1 is relatively stronger on subject-specific tasks). These results highlight when $L_{GO}$ helps, precisely in settings where CE under-optimizes a semantically important token group, and show how grouping functions modulate the magnitude of the improvement.

---

### Official Review · Reviewer_haJz · 2025-10-31

**Soundness:** 3
**Presentation:** 2
**Contribution:** 3
**Rating:** 6
**Confidence:** 3

**Summary:**

The authors introduce SFT-GO, a supervised fine-tuning with group optimization, where they put the emphasis on token level selection rather than data instance selection like on other sampling methods introduced in the literature for SFT. They correctly highlight the gap in recent works in the literature that not enough in depth research has been conducted on token level sampling optimizations for
SFT.

They make two main claims: (1) minimizing our objective improves the accuracy of the model in all token groups compared to standard training, and (2) a simple mini-batch procedure preserves the O(1/√T) convergence rate and converges to the global optimum. Their methodology to show this focuses on a) a statistic based method using TF-IDF scores, b) a semantics based method using token selection probabilities, and c) a task-specific approach using an external model. Their evaluation focuses on two main datasets, LIMA, and Alpaca.

The main contribution of the paper is introducing the analytical foundation of their group optimization setup for token selection. They define a group function and use that to define a new loss function which is a combination classical avg loss over all tokens + a worst case scenario of loss over tokens picker or not picked by the grouping function. See equations 1-4. Then they introduce two propositions to prove achieving better loss, an better convergence time.

Overall, the paper is well written, with their main contribution being on the analytical aspects of the group optimization. They

**Strengths:**

- The paper provides compelling empirical evidence that standard supervised fine-tuning under-optimizes semantically important tokens relative to common functional tokens, motivating the need for differential treatment across tokens.

- The paper is strong in its mathematical rigor and analytical proof. It successfully proves the two propositions, with fairly high complication in the proof process utilizing algebraic massages of probabilistic values and inequalities. Proposition 2 utilizes Jensen inequality successfully derive an error bound.

- The authors demonstrate that the method generalizes across several grouping strategies (TF-IDF, LLMLingua-2, and Rho-1), highlighting that the framework is not tied to a specific token importance estimator.

**Weaknesses:**

- The choice of token importance threshold 𝜂 meaningfully affects performance, and the paper shows non-monotonic behavior as this parameter varies. This suggests additional tuning is required.

- The method’s effectiveness varies significantly across grouping strategies. For example, LLMLingua-2 performs better than TF-IDF, reflecting dependence on access to an external semantic model. Thus, SFT-GO’s benefits are not inherent—they rely on choosing a strong importance estimator.

- Although the paper motivates why token level optimization is needed, it still lacks enough comparison to other forms of data sampling. Other techniques include sampling methods using optimal design, information gain importance sampling, etc.

**Questions:**

The authors provide a justification for their theoretical assumptions in Appendix C. It would be great if I can learn more on what other challenges were preventing them to lift these restrictions and prove more general results with less restrictive assumptions.

---

> ### Author Response · Authors · 2025-11-25
> **Rebuttal by Authors**
>
> > Weakness 1 (W1): The choice of token importance threshold 𝜂 meaningfully affects performance, and the paper shows non-monotonic behavior as this parameter varies. This suggests additional tuning is required.
>
> $\eta \in (0, 1)$ is a hyper-parameter that determines whether a token is considered important for the grouping function $g$. A higher value of $\eta$ results in more tokens being classified as unimportant. Figure 3 in the ablation study shows that all SFT-GO methods achieve the best performance when $\eta$ is between 0.90 and 0.55. Moreover, across this range, all SFT-GO methods consistently outperform the base-SFT method, demonstrating the robustness of SFT-GO to different $\eta$ values.
>
> > W2: The method’s effectiveness varies significantly across grouping strategies. For example, LLMLingua-2 performs better than TF-IDF, reflecting dependence on access to an external semantic model. Thus, SFT-GO’s benefits are not inherent—they rely on choosing a strong importance estimator.
>
> We agree that different token-importance estimators vary in effectiveness, and this is expected: SFT-GO is designed as a general framework that accepts any token-importance prior, rather than committing to a single estimator. The goal of the paper is not to show that one grouping strategy is universally superior, but to demonstrate that injecting token-importance priors into SFT, regardless of the specific estimator, provides consistent improvements.
>
> As shown in our experiments, all three representative grouping strategies outperform the standard SFT baseline across eight benchmarks, despite their differing strengths. This indicates that the value of SFT-GO lies in the framework itself, which can benefit from stronger importance estimators as they develop, rather than relying on a particular one.
>
> > W3: Although the paper motivates why token level optimization is needed, it still lacks enough comparison to other forms of data sampling. Other techniques include sampling methods using optimal design, information gain importance sampling, etc.
>
> Thank you for the suggestion. Our method leverages token-level importance in supervised fine-tuning and is complementary to sampling-based methods. Following your suggestion, we conducted additional experiments using the popular sampling method [3]. Specifically, we sampled the top 40% of LIMA using [3].
>
> | Metric         |        |
> |----------------|------------
> | Base-SFT         | 40.55 |
> | LESS [3]        | 41.36 |
> | LLMLingua    | 41.95 |
> | LLMLingua + LESS      | 42.46 |
>
> The table shows that the sampling method (LESS) achieves better performance than Base-SFT when fine-tuned on the selected samples, while LLMLingua still outperforms LESS. Moreover, when the token-prior method is combined with the complementary sampling method, we obtain even better performance. We have added Appendix I in the revised paper, and discussed these results.
>
> [3] LESS: Selecting Influential Data for Targeted Instruction Tuning
>
> > Question 1 (Q1): The authors provide a justification for their theoretical assumptions in Appendix C. It would be great if I can learn more on what other challenges were preventing them to lift these restrictions and prove more general results with less restrictive assumptions.
>
> Thank you for the helpful suggestion. In Appendix C, we justified the practical assumptions used in our analysis: bounded domain, Lipschitz continuous subgradients for $L_{GO}$, and convexity. Below we clarify the challenges that arise when attempting to remove these assumptions.
>
> - Bounded domain: Without a bounded parameter space, controlling gradient behavior becomes difficult and typically requires more structural assumptions on the model or data (e.g., bounded logits), which are less realistic than the effective norm bounds already enforced in modern training.
> - Lipschitz subgradients: Dropping this assumption breaks the standard tools used to control worst-case group deviations. Extending these bounds to non-Lipschitz settings remains an open challenge in DRO theory, as the loss can vary too sharply to guarantee stable analytical control without imposing significantly stronger regularity conditions.
> - Convexity: Removing convexity eliminates several well-established techniques for bounding the worst-group objective and analyzing its stability. In non-convex settings, the worst-group loss can significantly exhibit complexity, making the theoretical analysis more intricate and preventing clean, assumption-light guarantees. Convex convergence results, which are widely used to analyze neural networks, offer valuable insights into the stability of non-convex models.

---

### Official Review · Reviewer_C5zo · 2025-11-01

**Soundness:** 3
**Presentation:** 3
**Contribution:** 2
**Rating:** 6
**Confidence:** 2

**Summary:**

SFT-GO is an updated supervised fine-tuning algorithm for LLMs where the authors propose an importance-based token-grouping instead of treating all tokens equally. This grouping based on a weighted combination of the standard cross-entrpopy loss and a worst-group loss. This encourages the model to perform well on rare but semantically rich tokens as well. Empirical results on the Alpaca and LIMA benchmarks using the llama 3.1 8B and 3.2 3B models shows a consistent improvement over 7 reasoning and QA benchmarks.

**Strengths:**

1. Connecting group DRO to LLM finetuning is simple and intuitive to understand. An added benefit that can be expected is indirectly reducing token-level spurious correlations (aka reliance on filler words, etc.).
2. The empirical results across different LLM benchmarks show an improvement in average performance for minimal change in training setup.
3. The definition of the grouping function is interesting - particularly the use of llmlingua 2 for semantics-based grouping and its effectiveness.
4. A study of token importance during training could be a whole paper in itself - a research field that could deserve some attention. Particularly interesting when it comes to multi-language or multi-modal training.

**Weaknesses:**

1.  The dependence on tools like llmlingua for grouping can create a suboptimal dependency on certain domains (for instance the drop in performance in Math QA in Table 2). As mentioned by the authors, any input biases in these models or the training data (in case of TF IDF) will be amplified in training.
2.  Qualitative examples of the groups and semantically-rich tokens determined by their algorithm would make it easier to support these otherwise intuitive claims.
3. While few works consider token-level importance, there are lots of sample-level data valuation frameworks [1] [2] . This paper could have used a baseline to compare how SFT-GO compares to these methods. The correlation (or lack of) between high-valued samples and high valued tokens would be interesting to note.


[1] What is Your Data Worth to GPT? LLM-Scale Data Valuation with Influence Function
[2] GREATS: Online Selection of High-Quality Data for LLM Training in Every Iteration

**Questions:**

1. Can the authors offer an experiment to prove the convergence rates are the same as standard SGD under convex assumptions, as claimed in the theoretical analysis.
2. Impact of SFT-GO on downstream preference-optimization/ alignment tasks is unclear - can the authors comment on the same?
3. An interesting future experiment could be the interaction between SFT-GOs important token groups and attention saliency maps.

---

> ### Author Response · Authors · 2025-11-25
> **Rebuttal by Authors**
>
> > Weakness 1 (W1): The dependence on tools like llmlingua for grouping can create a suboptimal dependency on certain domains (for instance the drop in performance in Math QA in Table 2). As mentioned by the authors, any input biases in these models or the training data (in case of TF IDF) will be amplified in training.
>
> SFT-GO is a general optimization framework and is intentionally agnostic to how token groups are constructed. Practitioners can swap out LLMLingua or TF-IDF for domain-appropriate scoring functions (e.g., math-specific saliency models.) SFT-GO neither requires nor assumes any particular grouping prior; rather, it provides a unified mechanism for leveraging any such prior to rebalance optimization.
>
> We also note that the framework can mitigate the risk of bias amplification: because SFT-GO explicitly applies optimization pressure to high-loss groups, it is often more robust to imperfect priors than methods that rely solely on those scores. Exploring domain-specialized grouping functions is an exciting future direction, but we emphasize that the framework itself remains flexible and domain-independent.
>
> > W2: Qualitative examples of the groups and semantically-rich tokens determined by their algorithm would make it easier to support these otherwise intuitive claims.
>
> Thank you for your thoughtful suggestion. We provide an analysis of token-importance assignments by each grouping function in Appendix H. There, we include qualitative examples that illustrate the types of tokens each method considers important.
>
> LLMLingua-2 highlights syntactic connectors such as "but" and "although," reflecting its sensitivity to sentence flow and linguistic cohesion rather than purely content-bearing terms. TF-IDF often ranks infrequent terms but also overweights common function words, underscoring its frequency-driven nature. In contrast, Rho-1 consistently surfaces semantically meaningful, domain-specific tokens such as "buns" and "temperature," aligning with its focus on excess loss. These examples illustrate the distinct principles underlying each method’s token-selection behavior and provide concrete support for the claims made in the main text.
>
> > W3: While few works consider token-level importance, there are lots of sample-level data valuation frameworks [1] [2] . This paper could have used a baseline to compare how SFT-GO compares to these methods. The correlation (or lack of) between high-valued samples and high valued tokens would be interesting to note.
>
> Thank you for the suggestion. Our method leverages token-level importance in supervised fine-tuning and is complementary to sampling-based methods. Following your suggestion, we conducted additional experiments using the popular sampling method [3]. Specifically, we sampled the top 40% of LIMA using [3].
>
> | Metric         |        |
> |----------------|------------
> | Base-SFT         | 40.55 |
> | LESS [3]        | 41.36 |
> | LLMLingua    | 41.95 |
> | LLMLingua + LESS      | 42.46 |
>
> The table shows that the sampling method (LESS) achieves better performance than Base-SFT when fine-tuned on the selected samples, while LLMLingua still outperforms LESS. Moreover, when the token-prior method is combined with the complementary sampling method, we obtain even better performance. We have added Appendix I in the revised paper, discussed these results, and cited the suggested papers. Thank you.
>
> [3] LESS: Selecting Influential Data for Targeted Instruction Tuning
>
> > Question 1 (Q1): Can the authors offer an experiment to prove the convergence rates are the same as standard SGD under convex assumptions, as claimed in the theoretical analysis.
>
> Following your suggestion, we performed additional experiment. To empirically validate Proposition 2, we constructed a fully convex logistic-regression setting with a fixed grouping function and ran minibatch SGD. We included Figures 5 in Appendix H. Figure 5(a) plots the excess error $L(\bar{\theta}_T) - L(\theta^*)$ in log-log scale.
>
> The CE and $L_{GO}$ curves almost perfectly overlap and decay along a straight line, consistent with the expected $1 / \sqrt{T}$ rate. Figure 5(b) plots the rescaled quantity $\sqrt{T}[L(\bar{\theta}_T) - L(\theta^*)]$; both methods produce nearly flat curves over the entire optimization horizon, which is the characteristic signature of $O(1/\sqrt{T})$ convergence. These results empirically confirm the convergence rate of Proposition 2 in a convex regime.

---

> ### Author Response · Authors · 2025-11-25
> **Rebuttal by Authors**
>
> > Q2: Impact of SFT-GO on downstream preference-optimization/ alignment tasks is unclear - can the authors comment on the same?
>
> The token-level grouping mechanism of SFT-GO offers a complementary benefit for downstream preference-optimization methods such as DPO or PPO. Since preference alignment operates on sequence-level log-probabilities that decompose naturally into token-level contributions, strengthening the model's likelihoods on different token groups leads directly to more stable preference gradients and lower variance during alignment. As a result, the improvements introduced by SFT-GO at the token level carry forward into subsequent preference-optimization stages.
>
> > Q3: An interesting future experiment could be the interaction between SFT-GOs important token groups and attention saliency maps.
>
> Thank you for your suggestion. We agree that the interaction between SFT-GO and attention saliency maps could be interesting. As this would require substantial additional analysis, we leave it to future work. Thank you.

---

### Author Response · Authors · 2025-12-03
**Message to AC**

We thank the AC and reviewers for their time and constructive feedback.

SFT-GO is a simple, modular group-DRO–style framework that incorporates token-importance priors into supervised fine-tuning. It supports multiple grouping strategies (LLMLingua-2, TF–IDF, Rho-1), maintains standard SGD convergence, and consistently improves over baseline SFT across diverse benchmarks and model families (LLaMA-3.x, Qwen-2.5) and sizes (1.5B, 3B, 8B, and 14B).

### **Strengths Noted**

- **Conceptual clarity**: Reviewers emphasized that SFT-GO is intuitive, easy to integrate, and clearly presented (*C5zo*, *Ha2G*, *r4vC*).

 - **Strong motivation**: The persistent imbalance between important and unimportant tokens was viewed as a compelling and well-supported motivation (*C5zo*, *haJz*, *r4vC*).

 - **Theoretical rigor**: The analysis of Propositions 1 and 2 and the connection to Rho-1 as a special case were recognized as sound and clearly explained (*haJz*, *r4vC*).

 - **Framework generality**: SFT-GO’s compatibility with different token-importance estimators was consistently highlighted (*C5zo*, *haJz*, *r4vC*).

 - **Practicality and reproducibility**): The method requires minimal changes to existing SFT pipelines, has light hyperparameters, and includes solid ablations (*Ha2G*, *r4vC*).

 - **Significance and potential**: Reviewers noted that token-level optimization is underexplored and that SFT-GO provides a flexible base for future extensions (*C5zo*, *r4vC*).

### **Addressing Reviewer Concerns**

- *C5zo*: We clarified SFT-GO’s modularity and highlighted qualitative examples of token groups in the appendix. We incorporated a comparison with LESS (a sampling-based data-selection method) to show that token-level and sample-level approaches are complementary. We added a convex logistic-regression to empirically validate the theoretical convergence behavior as well.

 - *haJz*: We emphasized the robustness of SFT-GO to the importance threshold
$\eta$, as shown in the original ablations, and clarified that the performance differences from SFT-GO variants stem from the chosen importance estimator, not the framework. We compared with LESS (a sampling-based data-selection method) to show that SFT-GO is complementary. We also explained why our theoretical assumptions were practical and necessary.

 - *Ha2G*: We conducted statistical significance testing, and the p-values confirm that SFT-GO yields significant performance gains. We further extended our experiments to another model family (Qwen2.5) and additional sizes (1.5B and 14B). We showed that our objective function outperforms standard CE through empirical analyses, extensive experiments and ablations, and mathematical proofs. The assumptions behind our theorems hold in practice, as demonstrated in the rebuttal. We also clarified the confusion around Fig. 1(b) and strengthened the motivation.

 - *r4vC*: We reinforced our ablation study on the hyper-parameter $\lambda$ and justified both our analysis and the chosen hyper-parameter setting. We also clarified the implication of Proposition 1 and its generalization guarantee under the same train–test distribution, which is a standard assumption in loss analysis. Finally, we strengthened the motivation by clarifying Fig. 1(b).

Overall, SFT-GO provides a principled, token-aware alternative to standard SFT; grounded in theory, simple to adopt, and consistently effective across models, datasets, and grouping functions. We have addressed all reviewers’ comments and incorporated their suggestions into the revision.

Regards,

Authors

---

### Meta-Review · Area_Chair_g8oq · 2026-01-07

**Summary:**

This paper proposes SFT-GO, a supervised fine-tuning objective that augments standard cross-entropy with a worst-case (Group-DRO–style) token-group loss, where token groups are defined by external or heuristic importance estimators (e.g., TF-IDF, LLMLingua-2, Rho-1). Reviewers generally found the formulation simple, modular, and easy to integrate, and agreed that token-level imbalance during SFT is a real and under-explored issue. However, the core concern driving the decision is that while the worst-case optimization perspective is the most interesting conceptual aspect, the paper largely introduces this objective directly rather than analyzing it in depth in the context of token-level learning. The method primarily instantiates a standard Group-DRO objective at the token level, with theoretical results that rely on generic convex/Lipschitz/bounded-domain assumptions and do not yield new insights specific to LLM SFT or token-group dynamics. Empirical improvements are generally modest and incremental, and the effectiveness of the approach is highly dependent on the choice and quality of the token-importance estimator, which limits the conceptual contribution. Overall, while the paper is well written and technically sound, the contribution is best characterized as incremental, reframing existing worst-case optimization ideas for token-level fine-tuning without sufficiently deep analysis or novel theoretical insight to justify acceptance.

**Reviewer Concerns:**

### Addressed by the rebuttal (partially to substantially):

- “Gains may be within noise” / request for significance tests (Ha2G): authors added significance testing and expanded experiments to additional model family/sizes (per rebuttal). This helps with robustness of the reported deltas, though it does not change the underlying magnitude/novelty concern.

- Need for comparisons to sample-level data selection (C5zo, haJz): authors added a LESS-style baseline/combination (per rebuttal), which strengthens positioning as “token-level is complementary to sample-level.”

- Convergence-rate sanity check under convex assumptions (C5zo): authors added a convex logistic regression experiment (per rebuttal), which is responsive to the specific request, though it remains a stylized validation rather than addressing non-convex LM dynamics.

### Still outstanding:

- Novelty / conceptual contribution: Even with clearer framing, the core objective is a straightforward token-level application of worst-group optimization.

- The work reads as an incremental repackaging of known DRO/group-loss ideas plus “plug-in” importance estimators, and the “Rho-1 as a special case” framing further reinforces incremental novelty.

- Dependence on the grouping prior / external tooling: The paper’s effectiveness varies with the estimator (LLMLingua-2 vs TF-IDF vs Rho-1), and the approach can inherit biases or domain mismatch from the grouping model; this is an intrinsic limitation of the framework as presented .

- The rebuttal clarifies “population vs empirical” language, but the theoretical development remains largely non-specific to LLM SFT and does not provide a new mechanism-level explanation for when the token worst-group term should help beyond the general DRO intuition.

- Objective mechanics / stability near argmax switches: The rebuttal describes implementation behavior, but the paper itself still could be clearer about practical switching frequency, tie handling, and how this interacts with token partition noise; this affects reproducibility/interpretability for practitioners.

**Reviewer Scores:**

I don't think it will change any scores like from (2 to 4 or 4 to 6).

---

### Decision · Program_Chairs · 2026-01-26

Reject